# Emergence of Phonemic, Syntactic, and Semantic Representations in Artificial Neural Networks

## Abstract

During language acquisition, children successively learn to categorize phonemes, identify words, and combine them with syntax to form new meaning. While the development of this behavior is well characterized, we still lack a unifying computational framework to explain its underlying neural representations. Here, we investigate whether and when phonemic, lexical and syntactic representations emerge in the activations of artificial neural networks during their training. Our results show that both speech- and text-based models follow a sequence of learning stages: during training, their neural activations successively build subspaces, where the geometry of the neural activations represents phonemic, lexical, and syntactic structure. While this developmental trajectory qualitatively relates to children's, it is quantitatively different: These algorithms indeed require 2 to 4 orders of magnitude more data for these neural representations to emerge. Together, these results show conditions under which major stages of language acquisition spontaneously emerge, and hence delineate a promising path to understand the computations underpinning language acquisition.

## 1 Introduction

To acquire language, humans follow a remarkably stereotyped developmental trajectory: infants first learn to distinguish phonological categories (Kohl, 1993; Eilers et al., 1979; Jusczyk et al., 1993; Kuhl, 2004), and only later become able to understand and produce complex semantic and syntactic structures (Gleitman, 1990; Pinker, 1996; Christophe et al., 2008; Gomez & Gerken, 1999; Friedmann & Rusou, 2015; Boudreault & Mayberry, 2006). Linguistic theory has long posited the existence of distinct structures for these various domains of language – for example, nested-trees for syntax, hierarchical graphs for lexical semantics, and the vowel trapezoid for phonemes. While the acquisition of such structures is increasingly well characterized (Friedmann et al., 2021; Dupoux, 2018), the underlying neural and computational mechanisms of language acquisition remain poorly understood (Kuhl, 2010; Frank, 2011; Martin et al., 2022; Kachergis et al., 2022; Dubois et al., 2016; Evanson et al., 2025).

Correlates of these structures were found to organize large language models (LLMs) activations, and consequently, LLMs appeared as a fruitful path to propose a mechanistic theory of language acquisition (Frank, 2023; Vong et al., 2024; Dupoux, 2018). Specifically, the activation patterns of self-supervised models represent simple semantic relationships (Landauer & Dumais, 1997; Jawahar et al., 2019; Pasad et al., 2021; 2024a; Park et al., 2025), syntactic structure (Hewitt & Manning, 2019; Tenney et al., 2019; Evanson et al., 2023) and phonological categories (Baevski et al., 2020a; De Heer Kloots & Zuidema, 2024).

However, it is unknown whether modern LLMs learn to instantiate different linguistic structures using a common principle. We suggest that this gap of knowledge is due (1) to the opacity of neural networks whose representations instantiate simultaneously all linguistic structures, and (2) to the lack of longitudinal measurement of these representations during training. To fill this gap, we study the geometry of these structures in neural space and the order in which they emerge in the models during training, much like in human language acquisition. We generalize (Hewitt & Manning, 2019)'s structural probe to test whether and when the representations of phonemes, lexical semantics

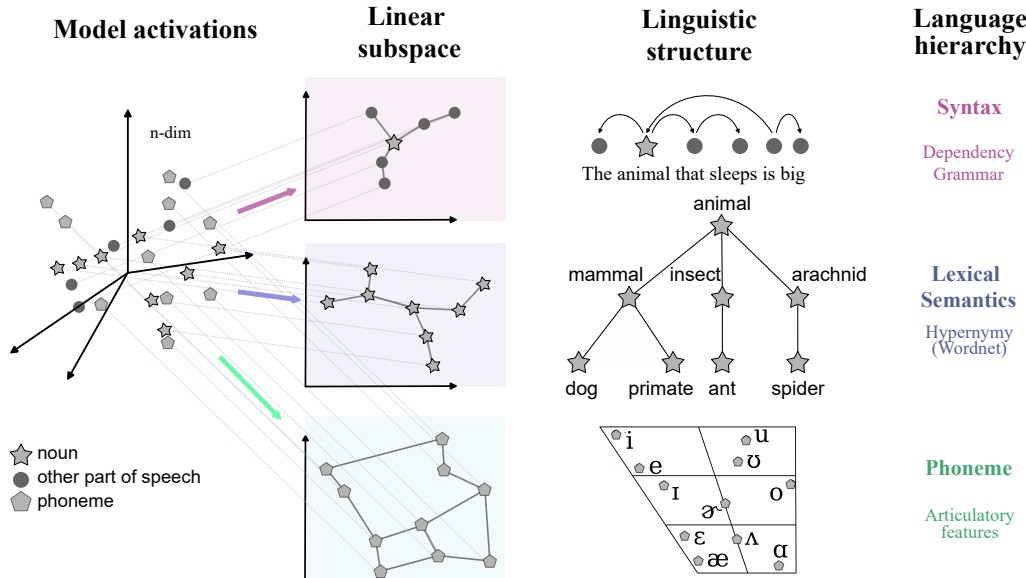

Figure 1: **Linear probing recovers phonemic, lexical semantic, and syntactic structure from the activation spaces of neural networks trained via Self-Supervised Learning.** The networks' neural activations live on a high-dimensional space in which different linear subspaces coexist. Different linear probes recover linear subspaces which represent phonemic, lexical semantic and syntactic structures, as long postulated by linguistic theories.

and syntactic trees emerge during the training of speech and text models. More precisely, we test the hypothesis that these linguistic structures are neurally encoded using a geometric code. We evaluate whether there exists a subspace of activations in which distances between model activations linearly correspond to the phonemic, lexico-semantic, or syntactic distances predicted by linguistics.

## 2 METHODS

### 2.1 A SHARED LINEAR PROBE FOR EXTRACTING LINGUISTIC STRUCTURES FROM MODEL ACTIVATIONS.

**Problem.** Linguistics predicts that language inputs (e.g. text, speech) should be represented with different levels of representations (e.g. phoneme, words, syntax). Each level of representation can be formalized as a metric system, which indicates e.g. whether different phonemes are more or less similar to one-another or whether different words in a sentence are more or less syntactically related. Formally, this prediction implies that language representations reduce down to a distance matrix. Here, we aim to evaluate whether language models learn to build subspaces in which activation pattern distances correlate with these distance matrices. For this, we extend the use of the *Structural Probe* (Hewitt & Manning, 2019), originally restricted to syntax, to any language representation, and fit this linear probe to identify subspaces that represent phonemic, lexical semantic, and syntactic features. Following this established interpretability paradigm, we restrict our focus to probing of representations can be directly used by downstream neurons through a simple linear readout. In contrast, non-linear probing could recover the target distances from arbitrary representations, thus hindering interpretability.

**Objective.** Consider a set of activation vectors of dimension $k$ in response to $n$ elements (e.g. word, token, phonemes): $h \in \mathbb{R}^{n,k}$. Given a pairs of elements $(i, j)$ in the set of possible pairs $S$, and a target distance $d_{i,j} \in \mathbb{R}^+$, the *Structural Probe* aims to find a linear transformation $B \in \mathbb{R}^{k,p}$ to extract the subspace where the distance between the elements' activations predict their target

distance (Hewitt & Manning, 2019).

$$\hat{B} = argmin_{B \in \mathbb{R}^{k,p}} \sum_{(i,j) \in S} |||(h_i - h_j)B||_2^2 - d_{i,k}|$$

Following Hewitt & Manning (2019), we use the squared Euclidean $||\cdot - \cdot||_2^2$ distance in the projected space. We optimize 2D ($B \in \mathbb{R}^{k,2}$) probe for visualization and 200D ($B \in \mathbb{R}^{k,200}$) probe for evaluation.

**Evaluation.** For consistency with previous works Hewitt & Manning (2019); Diego Simon et al. (2024); Limisiewicz & Mareček (2021), we evaluate a probe on each level of representation (Phoneme, Lexical and Syntax) using a Spearman correlation between true distances (proposed by linguistics) and those measured in the probe space.

## 2.2 PROBING DATASETS

We construct a paired text-speech benchmark made of different text datasets and their artificially synthesized speech counterparts (Table A in appendix).

**Syntactic representations.** The Universal Dependencies English Web Tree Bank (UD–EWT) corpus (Silveira et al., 2014) contains 16,622 sentences extracted from different web media. For each sentence, the syntactic tree is manually annotated by linguists according to the Universal Dependencies formalism (de Marneffe et al., 2021; Nivre et al., 2016), a type of Dependency Grammar (Tesnière, 1959). The UD–EWT corpus has been widely adopted to train both syntactic probes (Müller-Eberstein et al., 2022; Diego-Simón et al., 2024; Limisiewicz & Mareček, 2021) and syntactic parsers (Grünewald et al., 2021; Hershcovich et al., 2018; Yuan et al., 2019). For consistency, we keep the original train, validation and test splits provided in UD–EWT. The golden distance between two words of a sentence was defined as the number of edges between each word in the dependency tree of this sentence.

**Lexical representations.** Lexical semantics is assessed as a hypernymy relationship (is a generic term of) between units of senses (synsets) associated to nouns, in the WordNet semantic graph. For example, the synset "animal.n.01" is a hypernym to "cat.n.01". To obtain a maximal coverage of WordNet nouns, we construct a dedicated word-level dataset, named *WordNet–Nouns*, that contains most noun synsets in WordNet, while avoiding major polysemic issues. This procedure is detailed in the appendix. We focused on the noun subgraph of WordNet because including several parts of speech (POS) categories can lead to a large semantic score just because of an ability to discriminate between POS, and so, mislead to the conclusion that probes have learnt semantic structure. This resulted in a total of 35,352 sets of words. For the speech modality, we construct *WordNet–Nouns–TTS* by synthesizing every word with `seamless-m4t-v2-large` (Barrault et al., 2025) and aligning the resulting waveforms with the Montreal Forced Aligner (McAuliffe et al., 2017). To define a semantic distance, we turned to path length along the WordNet gaph. The gold distance between synsets A and B counts the number of WordNet edges (hypernym/hyponym relations) linking one synset to another.

**Phonemic representations.** To create a speech counterpart to UD–EWT, we synthesize all the sentences in the UD–EWT corpus using the Text-To-Speech model `seamless-m4t-v2-large` (Barrault et al., 2025). We then perform forced alignment with the Montreal Forced Aligner (McAuliffe et al., 2017) to obtain word and phone-level time-stamps for every synthesized utterance. Phoneme symbols were converted from ARPABET to IPA using the panphon library (Mortensen et al., 2016). Consequently, the resulting speech corpus preserves the original UD–EWT syntactic annotations while enriching them with phoneme identity labels. To define the distances among the different English vowel phonemes, we used dissimilarities of articulatory feature vectors from Mortensen et al. (2016). The dissimilarity value between 2 phonemes is an integer ranging from 0 (most similar) to 12 (most different), which represents the number of non-shared articulatory features.

## 2.3 MODELS

**Text models.** We use large language models from the LLama2 and Pythia suite (Biderman et al., 2023). Pythia models were auto-regressively trained on English text (Gao et al., 2020), differing only in their parameter size. Within this suite, we use models ranging in size from 70M to 1.4B parameters, for which the intermediate training checkpoints are publicly available [1]. The two Llama 2 models, of size 7B and 13B, allowed us to score models with stronger linguistic scores Waldis et al. (2024), defining an upper-bound for what the smaller Pythia models could achieve.

**Speech models.** We use audio Wav2Vec 2.0 models (Baevski et al., 2020b) previously trained on English speech, French speech, environmental sounds, and music (Orhan et al., 2025; Parcollet et al., 2024). Wav2Vec 2.0 models are pretrained in a self-supervised fashion to reconstruct masked segments of their own latent speech representations, an approach that encourages context-aware acoustic features. However, it remains unclear whether this self-supervised objective also fosters higher-level syntactic and semantic capabilities.

## 3 RESULTS

### 3.1 STRUCTURAL PROBES RECOVER THE GEOMETRY OF PHONEMIC ARTICULATION.

Phonemes can be discriminated from the activations of self-supervised speech models (Baevski et al., 2020b; Ji et al., 2022; Seyssel et al., 2022; De Heer Kloots & Zuidema, 2024). It remains unclear if their code for phonemes is structured according to articulatory features. To measure this, structural probes are trained to identify a phoneme subspace that instantiates articulatory features.

**Visualization and scores across models' layers** We first visualized the representations of a Wav2Vec 2.0 model pretrained on English speech through a less powerful 2-dimensional probe, optimized on the UD-EWT-TTS dataset. Remarkably, the projection of the test set (Fig. 2A) represents vowels similarly to the classic vowel trapezium (Fig. 2B): vowel phonemes with a similar articulation tend to be closer in the phonemic subspace. To evaluate how well the model instantiates the phoneme structure, we then train probes with a higher dimensionality. We find that phonemes are best encoded in layers with a relative depth of 0.5 and 0.8 (Fig. 2C), achieving a Spearman correlation of 0.75. This phonemic score outperforms by a large margin the untrained, random, baseline (phonemic score of 0.2). This demonstrates that instantiating the phoneme structure is not performed trivially with random acoustic processing.

**Effect of model size.** Is this a pure effect of model training, or is it also dependent on the model size? We find that the phonemic score scales with model size: larger models achieve higher scores than smaller ones (Fig. 2D). Among English-trained models, Wav2Vec 2.0 large (315M parameters) outperforms base (94M), which in turn outperforms tiny (30M), each models implemented with 24, 12, and 3 Transformer layers, respectively. This highlights that, despite its apparent simplicity, extracting the phoneme structure is easier with more contextual processing layers.

**Controls: acoustic pretraining is not enough** To verify that the instantiation of phoneme structure was due to speech processing instead of generic acoustic processing, we compared models trained on different data. Phonemic scores depend strongly on the training data (Fig. 2D): models trained on English speech perform best, whereas those trained on French, music, or environmental sounds perform poorly. Note also, that the music and environmental sounds models are not completely naive to speech, with an estimated 8% of speech inputs in the environmental sound dataset (Orhan et al., 2025). Only the French large and base models outperform the `tiny` English model, which is expected given the model's limited size. These controls prove that the instantiation of phoneme structure results from the emergence of speech-specific processes.

**Emergence during pretraining** How is this representation of phonemes learned? We probe the English Wav2Vec 2.0 94M model across its training checkpoints. For simplicity, we focus on the layer with the best phonemic score, i.e., layer 9. Fig. 2E shows a gradual rise in this score with the

---

[1] https://huggingface.co/models?other=pythia

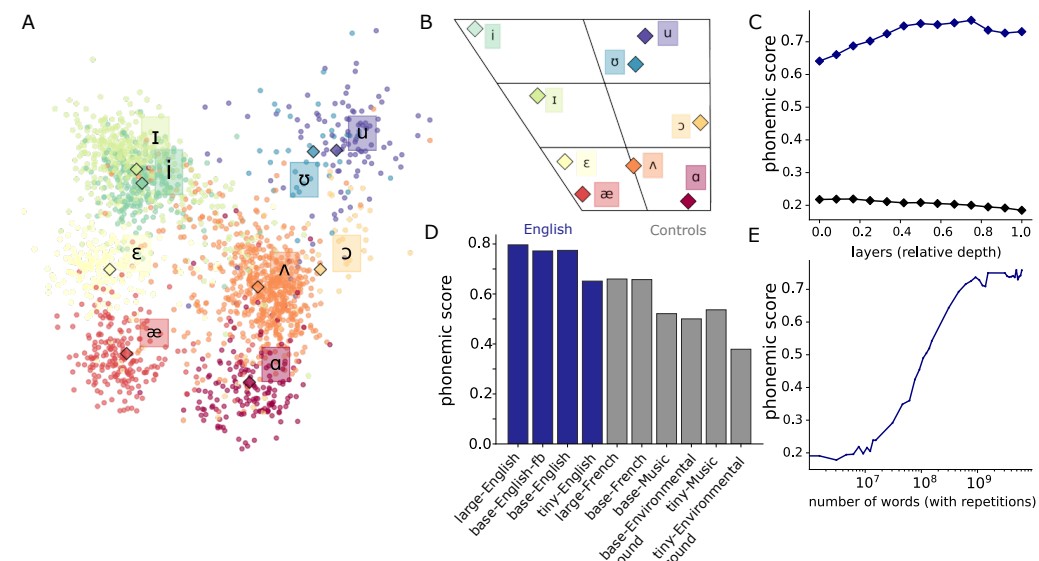

Figure 2: **Phoneme: structural probes recover the geometry of phonemic articulation. A.** Phoneme representations from a 2D structural probe trained to match pairwise articulatory distances among vowels. Responses to the same phoneme are colored identically. **B.** Canonical vowel trapezium indicating tongue height (vertical) and backness (horizontal). **C.** Layer-wise phonemic score for Wav2Vec2: probes trained on the final pretraining checkpoint peak near 80% relative depth and consistently outperform probes trained on the initial (untrained) checkpoint. **D.** Effect of model and data: English pretraining and larger model capacity both yield higher phonemic scores. **E.** Phonemic structure emerges gradually with pretraining, with scores increasing with the number of pretraining steps; the data required for Wav2Vec 2.0 to acquire robust phonemic structure far exceeds typical estimates of children's linguistic input.

cumulative spoken-word count, reaching emergence after roughly $10^9$ words (including repetitions), which corresponds to about 90,000 pretraining steps. This is considerably more than the $10^7$ words that children experience in their first year on average (Gilkerson et al., 2017). Taken together, these results indicate that self-supervised speech models spontaneously learn to instantiate a phoneme subspace whose geometry mirrors distances of articulatory features.

## 3.2 LEXICAL SEMANTICS

Language model representations have been known to encode semantic features (Korchinski et al., 2025). Additionally, models' behavior and the organization of their final layer have been shown to be consistent with an instantiation of lexical semantic relationship (Cohen et al., 2023), including the WordNet graph (Park et al., 2024). To measure the internal layers' representation, we train structural probes to predict the WordNet noun graph from text and speech models.

**Visualization and scores across models' layers.** We first optimized a 2-D probe of text models' semantic representations in order to visualize their representations. We display the training and test elements together while evaluating only across test elements. To our surprise, a 2-D probe already embeds well-constructed semantic graphs, especially when we focus on small subsets of the larger WordNet tree. As an example, we plot in Fig. 3A, a 2-D embedding of the tree below the mammal synset from the activations of a Pythia-1.4B text model. Other examples for several "supersense" semantic categories are available in Figs. E and F in appendix, as well as an example of 2-D embedding of the whole WordNet tree, see Fig. B in appendix. In two dimensions, this embedding does not capture the exact tree, but rather positions well the largest category of this semantic tree, as illustrated by plotting these categories' centroid in Fig. 3A-B (diamond markers). Smaller categories and the exact edges of the tree require a larger embedding space. We consequently train and evaluate a 200-D probe of the whole WordNet graph, in text and audio models. For this 200-D probe, the

semantic scores across model layers follow an inverted U-shape in all models: Wav2vec 2.0 (160M, max 0.1), Pythia (1.4B, max 0.29), and Llama2 (13B, max 0.41), Fig. 3C. This is remarkable, because transformer's tokens are vectorised and could therefore contain some semantic information. Here, we show that such semantic information is not sufficient to instantiate the lexical semantic structure of nouns. Instead, several computational steps are performed to merge and position these tokens' embedding together into a semantic subspace.

**Effect of model size.**   This questions the computational power necessary for these semantic spaces to emerge. We measured the maximal semantic score across layers as a function of parameters in a set of text and audio models, Fig. 3D. Remarkably, semantic scores increase linearly with text model sizes in the Pythia model suite and finally plateau at a large (7B) size for the Llama2 models. Similarly, the semantic scores of audio models increase with model size, although through a less steep slope. At equivalent size, text models have slightly larger semantic scores than audio models. This is in stark contrast with their very distinct training regime, with audio models seeing a completely distinct amount of data. This hints at a possible different training dynamic between audio and text models.

**Emergence during pretraining.**   We measured the evolution of the semantic score as a function of pretraining of a base Wav2vec 2.0 model (94M) trained on librispeech data, and a 1B Pythia model, Fig. 3E. Audio models' semantic scores initially increased faster than text models' scores, but then saturated. This saturation of the audio models questions whether they discovered semantic structure or if these scores could be explained by the acoustic properties of the words.

**Controls: semantic mostly arise from acoustic cues in audio models**   Acoustic cues could be sufficient to encode a small part of the semantic subgraph. For example, "dog" and "hunting dog" are expected to be close by in the graph, but this could be predicted purely from acoustic features; here, the fact that "dog" is present in both sets of words. To control for these acoustic features, we evaluated tiny (30M) and base (94M) Wav2vec 2.0 models trained on music, environmental sounds, and a large (315M) model trained on French speech. Each of these audio models reached significant semantic scores, and these scores were in the same range as the semantic scores of models exposed to English speech, but remained much lower than the scores of text models (Table D in appendix). With this structural probe, it consequently seems that most of the audio models' scores are due to acoustic correlates. Despite their low semantic score, the English-exposed models better clustered 89% of semantic categories of the WordNet tree than control models, with a 0.1 F1-score gain on average (Fig C in appendix, paired t-test $3.8 \times 10^{-36}$). Together, these results demonstrate a clear and modest but significant instantiation of lexical semantic structure by text and audio models, respectively.

### 3.3 SYNTAX

Language models are known to code syntactic trees through distances (Hewitt & Manning, 2019), while syntactic trees per se have yet to be found in speech models, since previous approach focused on decoding statistical quantities derived from the trees, like node depth (Pasad et al., 2024b). Here we train structural probes to measure if speech models also use a distance code to instantiate syntactic trees.

**Visualization and scores across models' layers.**   We first visualized the 2-D probing of a Wav2Vec 2.0 base (94M) model activations in response to 6 sentences with increasing hierarchical depth (Fig. 4A and B). As expected, the true (gold) and predicted (black) syntactic trees closely matched for simple sentences, but became increasingly more distant for harder, complex sentences. We then evaluated syntactic scores with probing in 200 dimensions. We observed strong syntactic scores for all models, including Wav2vec 2.0 models (94M parameters, max: 0.76), Pythia model (1.4B, max: 0.81), Llama2 model (13B, max: 0.82) (Fig. 4C).

**Effect of model size**   For text and audio models, syntactic performances first increase then plateau as a function of the number of parameters, Fig. 4D. Remarkably, unlike semantic scores, text models' performances quickly plateau with model size. This indicates that syntactic structure computa-

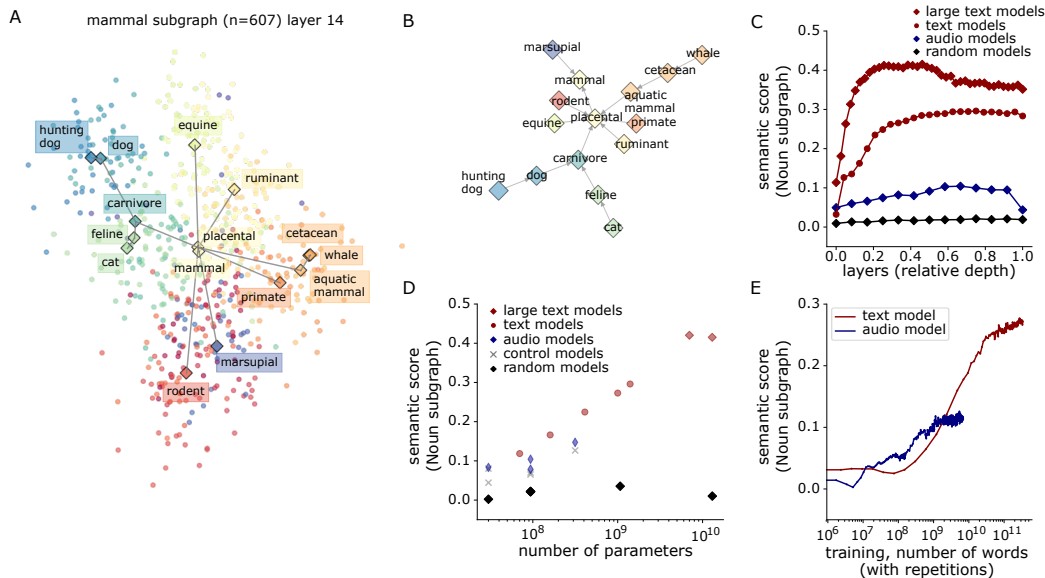

Figure 3: **Structural probes recover the geometry of lexical semantic graphs** A: 2d projection of text model (pythia-1.4B) response to all words in the mammal subgraph. B: Example WordNet subgraph, displaying synsets of the mammal subgraph with more than 50 hyponyms. c: Semantic score (for the graph composed of all nouns) for a large text (Llama-13B), text (pythia-1.4B), audio (Wav2vec 2.0-94M base), and random (Wav2Vec 2.0-94M base) models. D: Semantic scores for all models as a function of the model size. E: Semantic score as a function of the quantity of pertaining for the text (pythia-1.4B) and audio (Wav2vec 2.0-94M base) models.

tions do not require very large text models, but also that large text models do not develop a perfect syntactic parsing algorithm.

**Emergence during pretraining**   We plot in Fig. 4E the evolution of the syntactic score as a function of pretraining. Unlike text models, audio models' scores did not seem to saturate with the quantity of pretraining. For models of equivalent size, the emergence of syntax was faster in audio models, for which the dataset also used a smaller quantity of different sentences. This suggests that audio data provides syntactic cues that the model can discover and use to build syntactic subspaces.

**Controls: models instantiating syntax encode more than a linear tree**   Similarly to semantics, could syntax processing in audio models be explained by acoustic features? If indeed control audio models reached relatively high syntactic scores, we found that this came from the oversimplicity of the metric. Indeed, control audio models partially predicted the syntactic trees, purely from the fact that they could predict a linear tree (where words are linked to their neighbors). To demonstrate this, we computed the Spearman correlation with a linear tree and compared it to the correlation with the gold dependency tree. This correlation were similar in the case of control models but not for English-exposed models. Critically, this was not the case for English-exposed models whose projected representations correlated more with the syntactic than the linear structure. This demonstrates that true syntactic representations in response to English sentences are present in English-exposed models but not in models trained on French, music, or environmental sounds.

### 3.4   ORDER OF ACQUISITION

We next compare the learning dynamics of these phonemic, semantic, and syntactic codes in the audio model. To reveal these emergences, we repeated phonemic, syntactic, and semantic probing across checkpoints logarithmically spanning the pretraining of the models. We plot in Fig. 5A-C the probing of 3 checkpoints that highlight the successive emergence of phonemic and lexical-syntactic structures in the audio models. We plot in Fig. 5D the emergence of these scores and fit a parametric curve for each. This allows us to then plot in Fig. 5E a relative score for each curve and

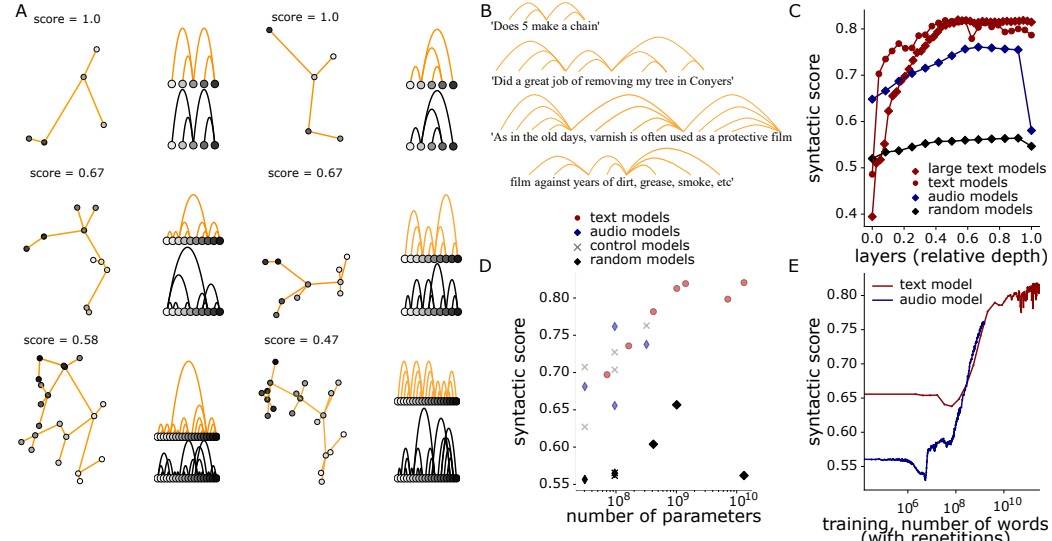

Figure 4: **Structural probes recover the geometry of syntactic trees** A. In each left subpanel, we plot an example of a 2d probe projection of the model activations, with trees reconstructed through the minimum spanning tree algorithm. On the right subpanel, we plot the gold (yellow) and predicted (black). B. Example of sentences and their syntactic trees. C. Syntactic scores of a large text (Llama-13B), a text (pythia-1B), speech (Wav2vec 2.0-94M base) and random (Wav2vec 2.0-94M base) models for each transformer layer. D. Highest syntactic score across layer for each model. E. Emergence of the syntactic abilities during pretraining, which includes several repetitions of the same dataset for the audio model.

compare their dynamic. Remarkably, we observe that phonemic emergence is followed by a partial emergence of lexical semantic and finally an emergence of syntactic abilities in the audio models. These emergences are each roughly separated by an order of magnitude of input data, indicating that each stage solidifies before the previous one. Yet, as a semantic code only partially emerges in the audio model, it remains to be seen if these models truly predict that lexical semantic structure appears before syntactic structure.

## 4 DISCUSSION

**Goal and approach**   To evaluate whether modern language algorithms can help understand the computational underpinning of language acquisition, we generalized Hewitt & Manning (2019)'s Structural Probe and assessed the emergence of phonemic, lexical, and syntactic representations during the training audio and text models.

**Similarities to Human Language Acquisition.**   In spite of their remarkably distinct architecture and learning principles, audio models appear, like children, to spontaneously build linguistic features, in a developmental trajectory that, at least at a coarse view, follows a phonemic and lexico-syntactic sequence. This emergence effectively captures some of the developmental trajectory of children, who – during their first year of life – demonstrate a progressive categorization of phonemes, and subsequently develop an ability to understand and produce syntactic structures from an increasingly large and precise vocabulary (Gomez & Gerken, 1999; Kuhl, 2004; Dupoux, 2018; Friedmann & Rusou, 2015; Boudreault & Mayberry, 2006; Sainburg et al., 2022).

**A quantitative gap.**   Consistent with previous work (Dupoux, 2018; Lavechin et al., 2022; Frank, 2011; Hu et al., 2024), data efficiency remains a major gap: compared to human children, the presently-studied speech and text models require 2-4 orders of magnitude more input data for these representations to emerge (Fig. 5 F) (Gilkerson et al., 2017). This discrepancy highlights that modern language algorithms remain remarkably inefficient, and thus calls for exploring novel neural

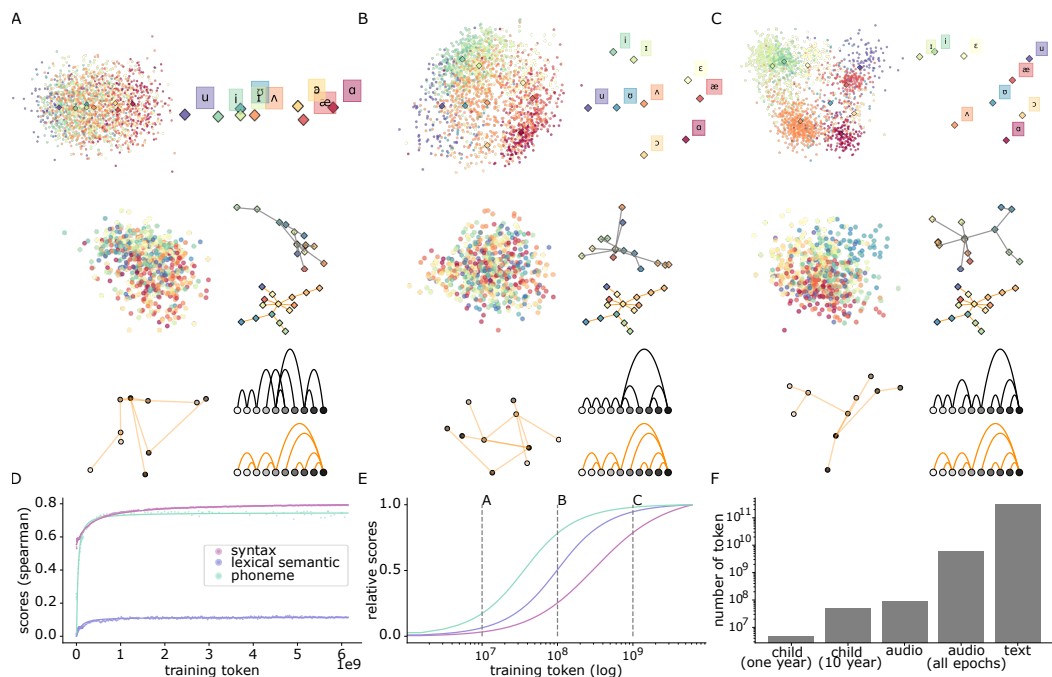

Figure 5: **Linguistic structures are acquired in sequential order** A, B and C: 2d visualizations of the phonemic and semantic space along with one example syntactic structure for 3 pertaining steps. D: Left: Emergence of semantic, syntactic, and phonemic score for a Wav2vec 2.0-94M base model (dots), along with a parametric fit (line) E: relative scores of the parametric fit (min to 0 and max to 1), demonstrating the successive emergence of phonemic and lexical-syntactic structures. Checkpoints used for the top sub-panel are highlighted with lines. F: average amount of words (tokens) heard by a child or used by models. Note that for the audio dataset, many epochs of optimization over the same dataset take place.

architectures and training paradigms (Wolf et al., 2023; Hu et al., 2024; Xiang et al., 2022). Part of this gap may be explained by non-language data pretraining. Indeed, Poli et al. (2024) showed that pretraining speech models on ambient sounds improves data-efficiency to then develop phonemic representations. Similarly, Orhan et al. (2025) showed that pretraining on ambient sounds spontaneously lead audio models to detect algebraic auditory structures.

**Modeling the developmental trajectory of representations.** This emergence strengthens earlier work on toy models. For example, (Kemp & Tenenbaum, 2008) showed that hierarchical concept structures can sequentially emerge from feature spaces by iteratively searching over a structured hypothesis space. More formally, Saxe et al. (2014; 2019) showed mathematically that hierarchical representations emerge sequentially in deep linear networks during training. Although we did not observe a sequential emergence of semantic categories (see Fig. A in appendix), we observe a sequential emergence across linguistic structures. Finally, Lavechin et al. (2025); Chen et al. (2024) show that self-supervised learning from raw speech and text yields a staged emergence of phonetic, early lexical structure and syntax without explicit labels. The present work complements these approaches by testing a unified code within self-supervised neural networks for the different linguistic structures posited by linguists.

**Limitations** Our findings come with several limitations. First, the developmental ordering between syntax and lexical semantics, a central question in cognitive science (Gleitman, 1990; Pinker, 1994), remains unresolved. Second, we report aggregate scores and do not stratify performance by simple versus complex syntactic phenomena or by token/phoneme frequency. Third, our evaluation relies exclusively on structural probing of internal representations, rather than on prediction-based tests at the phonemic, syntactic, or lexical levels (Schatz et al., 2013; Warstadt et al., 2020; Hill et al.,

2015; Nguyen et al., 2020). Thus, our conclusions concern representation rather than inference and may not fully characterize downstream performance.

Fourth, optimizing a probe to match pairwise distances might nevertheless be too restrictive. This approach might nevertheless be too restrictive. For example, the distance between "dog" and "river" does not have to be as precisely captured as the distance between "dog" and "cat". Additionally, the exact choice of distance, for example, the shortest path distance on the WordNet graph, is itself questionable. To show that our approach can be generalized beyond distance learning, while simultaneously strengthening our results, we present supplementary results using a "contrastive" probe, which optimizes the extraction of the topology of the structure, irrespective of the distance. This probe is optimized according to the contrastive objective of (Nickel & Kiela, 2017). On measures of topological fidelity (unlabeled undirected attachment score for syntax, rank for syntax and semantic), this probe achieved better scores than the distance probe, while revealing similar emergence dynamics (Fig. I in appendix). These results prove that our approach could, in future work, be extended by searching for different probing objectives.

Fifth, we do not probe phonemic structure in text models. This is because text models for which checkpoints were available used token-level inputs instead of letters or grapheme inputs.

Sixth, our analysis is restricted to nouns to avoid equating semantics and part-of-speech categorization. This analytical choice may limit the generality of the present finding. Future work should thus extend to other parts of speech and explore semantic relationships beyond hypernymy.

**Impact and Future Directions**  The present generalization of Hewitt & Manning (2019)'s structural probe to lexical semantics and phonology provides a unifying framework to characterize the structure and emergence of linguistic representations in neural networks. This approach reduces the notion of 'representation' to (squared) Euclidean distances between elements in specific subspaces of the activation space. This approach also provides a mechanistic explanation: As linear readout is a core operation of any neuron, linear subspaces make these language representations readily accessible to any subsequent steps (Kriegeskorte, 2015; Higgins et al., 2018; Park et al., 2024), including attention mechanisms. Our results, like previous interpretability works, suggest the existence of laws governing the geometry of neural networks' activations Elhage et al. (2022); Hernandez et al. (2024); Jha et al. (2025); Park et al. (2025); Landauer & Dumais (1997); Huh et al. (2024).

Our probing approach could be used to understand how structures are built through in-context learning. For example, (Orhan et al., 2025) show that algebraic auditory structures are detected by speech models through in-context learning. This work predicts that the code underlying this detection is a distance code which can be detected through a linear probing approach.

Perhaps most excitingly, this work delineates a path to model the brain bases of language acquisition (Di Liberto et al., 2023; Nakagi et al., 2025). In particular, Evanson et al. (2025) show that the representations learned by audio and text based models not only linearly align with those of the human brain (Huth et al., 2016; Caucheteux & King, 2022; Millet et al., 2022; Oota et al., 2023; Goldstein et al., 2022), but their developmental trajectory also captures the change of representations in the brain of children during their development. Overall, these findings suggest that modern language models are not only powerful tools for analyzing complex data but also playgrounds from which we can theorize and understand computational principles underlying language acquisition in the human brain.

## 5 REPRODUCIBILITY STATEMENT

We take the following steps to ensure reproducibility of this work: All models necessary to replicate this study can be downloaded from the HuggingFace repository, following Table C in appendix. All hyperparameters for every structural probe used in this study have been listed in Table B in appendix. All dataset generation and cross-validation procedures are presented in detail in the appendix. The codebase, which would allow to perform incremental work on top of this study, will be made available upon acceptance. Indeed, we spent a large amount of the work effort on engineering development and code optimization, including: rapid optimization of probes (bringing probe training times from days to under one hour, which enabled large grid search and probing across a large amount of checkpoints), implementation of a novel GPU technique for K-NN classification, and

extensive pipeline development for checkpoint storing. These developments only change the speed of our optimization pipeline without affecting its results.

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

# A   APPENDIX

| Name | Modality | Annotation |
|------|----------|------------|
| UD–EWT | text | UD (M) |
| WordNet–Nouns | text | WordNet (M) |
| UD–EWT–TTS | speech | UD (M) + Phonemes (A) |
| WordNet–Nouns–TTS | speech | WordNet (M) |

Table A: Datasets used to train syntactic, lexical semantic and phonological probes. Annotations of linguistic structures can be manual or automatic, marked by M and A respectively.

| Structure | Model | Learning rate | Dim | Epochs | Probe init | batch size |
|-----------|-------|---------------|-----|--------|------------|------------|
| phonemic | Wav2vec 2.0 | 0.00001 | 768 | 100 | 0.00001 | 1 set of 200 phonemes |
| semantic | Wav2vec 2.0 30-94M | 0.0000075 | 200 | 200 | 0.00001 | 300 set of 12 words |
| semantic | Wav2vec 2.0 315M | 0.0005 | 200 | 200 | 0.00001 | 300 set of 12 words |
| semantic | pythia | 0.0000025 | 200 | 300 | 0.00001 | 300 set of 12 words |
| semantic | Llama 2.0 | 0.00005 | 200 | 300 | 0.00001 | 300 set of 12 words |
| syntactic | Wav2vec 2.0 | 0.00001 | 200 | 2 | 0.00001 | 300 sentences |
| syntactic | pythia | 0.000025 | 500 | 1 | 0.00001 | 150 sentences |
| syntactic | Llama 2.0 | 0.00001 | 200 | 1 | 0.00001 | 300 sentences |

Table B: Parameters of the linear probe. All probes are trained with the AMSGrad variant (Reddi et al., 2018) of the Adam optimizer (Kingma & Ba, 2014) For each combination of (model, task), we perform a grid search across learning rate, spanning linearly 20 values from 0.0000001 to 0.01, and report the best learning rate. Unlike its sensitivity to learning rate, the probe is more robust to batch size and probe initialization value. The probe is randomly initialized from a uniform distribution between -0.00001 and 0.00001. For each model, all hyperparameters are constant across layers. When the model size is not given, it means that the same hyperparameters were used across all model sizes.

SYNTAX DATASET

Our generation pipeline will be made available in the form of the ALS (Auditory Linguistic Structure) python repository upon acceptance. This pipeline takes as inputs an annotated text corpus (ConLLu format), and synthesizes the equivalent auditory dataset, along with a precise temporal alignment. We make available a cluster-friendly version of this pipeline, which allows the reproduction of the dataset in less than one hour.

**Speech synthesis**   For each sentence in the English-Web-Treebank (EWT) Universal Dependencies Dataset, we synthesize a voice clip using the "seamless-m4t-v2-large" deep learning model. First, we remove from each sentence any remaining contractions using the contractions python package. Removing contractions allows the Force Aligner to separate in time contracted words, (like You're) that would be otherwise recognized as a single event. Second, for each sentence, the text is used as input to a "seamless-m4t-v2-large" model, with default English speaker voice. We save the generated sounds as a WAV file with a 16000 Hz sampling rate.

**Alignment**   Forced Alignment (FA) is a necessary step to know when each word in a sentence occurs in its corresponding synthesized voice clip. FA was conducted using the Montreal Forced Aligner (MFA) model, employing the english_us_arpa corpus and dictionary. To quantify the effect of different aligners, we also reproduced our analyses with the NeMo Forced Aligner. This aligner produced a worse alignment when manually checked, in accordance with Rousso et al. (2024) which compared MFA with other aligners and found MFA to outperform. Indeed, everything else fixed, the syntactic and semantic scores to the NeMo-aligned corpus were poorer, which indicates measurement error induced by the alignment. Despite the great care we took for this forced

| Training Dataset | Model Size | Training Steps | Checkpoints | Hugging face repository |
|---|---|---|---|---|
| Audioset | 30M | 100k steps | ✓ | NDEM/model-wav2vec2_type-tiny_data-audiosetfilter_version-0 |
| Audioset | 94M | 100k steps | ✓ | NDEM/model-wav2vec2_type-base_data-audiosetfilter_version-2 |
| FMA | 30M | 100k steps | ✓ | NDEM/model-wav2vec2_type-tiny_data-fma_version-0 |
| FMA | 94M | 100k steps | ✓ | NDEM/model-wav2vec2_type-base_data-fma_version-2 |
| LeBench | 315M | 200k steps | x | LeBenchmark/wav2vec2-FR-7K-large |
| Librispeech | 30M | 100k steps | ✓ | NDEM/model-wav2vec2_type-tiny_data-librispeech_version-0 |
| Librispeech | 94M | 100k steps | ✓ | NDEM/model-wav2vec2_type-base_data-librispeech_version-4 |
| Librispeech | 94M | 400k steps | ✓ | NDEM/model-wav2vec2_type-base_data-librispeech_version-2 |
| Librispeech | 94M | 400k steps | x | facebook/wav2vec2-base |
| Librispeech | 315M | 200k steps | x | facebook/wav2vec2-large |
| The Pile | 70M-1.4B | 143k steps | ✓ | EleutherAI/pythia-Xb |
| Meta Internal corpus | 7B | 500k steps | x | meta-llama/Llama-2-7b |
| Meta Internal corpus | 13B | 500k steps | x | meta-llama/Llama-2-13b |

Table C: Set of audio and text models along with their training dataset, the model size, total number of training steps, and availability of checkpoints. Control audio models were taken from the following paper: Orhan et al. (2025).

| Model | Training data | Semantic score (pearson correlation) |
|---|---|---|
| Wav2vec 2.0 30M | environmental sound | 0.044 |
| Wav2vec 2.0 94M | environmental sound | 0.065 |
| Wav2vec 2.0 30M | music | 0.080 |
| Wav2vec 2.0 94M | music | 0.070 |
| Wav2vec 2.0 315M | French | 0.126 |
| Wav2vec 2.0 30M | English | 0.08 |
| Wav2vec 2.0 94M | English | 0.104 |
| Wav2vec 2.0 315M | English | 0.147 |
| Pythia 70M | The Pile | 0.118 |
| Pythia 160M | The Pile | 0.166 |
| Pythia 410M | The Pile | 0.22 |
| Pythia 1B | The Pile | 0.27 |
| Pythia 1.4B | The Pile | 0.29 |
| Llama 2 7B | Meta Internal Corpus | 0.42 |
| Llama 2 13B | Meta Internal Corpus | 0.415 |

Table D: Scores of the semantic probe, (max pearson scores across layers)

alignment, all of our metrics remain a lower bound estimate of the true model's abilities, as there will necessarily be some alignment errors in the process.

The Montreal Forced Aligner does not map exactly each word in its input text to each annotated word in the input corpus. Some composite words can indeed be split by MFA. To avoid silent misalignment mistakes, we run a simple iterative merging algorithm to group set of split words. This procedure leaves us with a beginning and an end time for every annotated word in the corpus.

Finally, we correct the information of the input corpus (ConLLu file), in the case of contractions. Decontracted sets of words are kept split as several elements (this should be done already in the ConLLu), while for re-contracted word (as can't becoming cannot), we keep only the first ConLLu token of the set of tokens associated with the contractions.

SEMANTIC DATASET

**Semantic category**    In the context of WordNet, we defined a semantic category as the set composed of a synset and all synsets that can reach this synset through hypernymy relationship (Fig A in appendix). For example, the noun graph root is "entity.n.01", and therefore all elements in the graph are part of the category "entity.n.01". Similarly, all synsets below "mammal.n.01" are part of the "mammal" category, but "animal.n.01" which is a hypernym of "mammal.n.01" is not.

**Dataset generation**    The mapping between WordNet synsets and the set of nouns is neither surjective nor injective. Words are often polysemic, such that preserving words associated to a single synset would restrict our investigation to a small subgraph of the WordNet graph. Conversely, synsets are associated with multiple sets of words (lemmas). To deal with these two constraint, we can nevertheless take advantage of the fact that some synsets are over-precise and characterize a very particular use of the words. For example, time is associated to 15 synsets, among which: time.n.05: 'the continuum of experience in which events pass from the future through the present to the past', and time.n.02: 'an indefinite period (usually marked by specific attributes or activities)'.

Consequently, to generate a dataset of pairs (synset,lemma) from the WordNet graph, we applied the following heuristic. (function wordnet_unisemic_wordlist in the ALS package). First, we collected all synsets in the WordNet noun graph. Each synset is associated with a list of lemmas (each lemma is a word or multiple words). We then repeated the following procedure: Given this list of synset, we first, whenever it was possible, assigned to a synset all lemmas for which no other synset has this lemma in its list. We then removed from the list of lemmas associated to this synset all lemmas that were in the list of other synsets. This allowed to repeat the procedure, as some lemmas then became assigned to a unique synset. In short, this first procedure selects for all synsets the lemma that is most unique to this synset.

This procedure still left a large number of polysemic lemmas, each associated with multiple synsets. Among those synsets, there are leaf synsets which have no hyponyms and are therefore very precise sense. We consequently pruned these leaf synsets out (removing a total of 9593 leafs synsets). The rationale was that the non-contextual evaluation of the lexical semantics of a polysemic word should rather be on its general sense than its precise and context-specific sense. Finally, we preserved in the graph the synset whose remaining list had at least one lemma associated with only one synset. Given this subset of synsets and their associated list of unique lemmas, we ultimately selected the lemma that had the highest word frequency according to the wordfreq package.

**Cross-validation folds for Wordnet**    Next, we divided this large graph into train and test sets, using a splitting strategy that ensured that for each semantic category (defined as a synset and all its hyponyms), 20% of its members were in the test set. To be able to quantify how well each semantic category was coded in the model, we had to put in the test folds a subset of the words for each category. This assignment was not trivial, and we consequently came up with a simulated annealing procedure to obtain it. This procedure minimized a loss function that measures for all categories, the percentage of its elements in the test set. To be precise, the cost is defined as: $\sum_j \sum_{i \in C_j} (\delta_{i \in T} - 0.2|C_j|)^2$ where $\delta_{i \in T}$ is 1 if the synset i is in the test set, and $C_j$ is the set of element in the category j, and the first sum iterates across all categories. Running a simulated annealing procedure involves randomly changing an element to be part of the training or testing set. This change is kept if the cost decreases ($\Delta cost < 0$) or if we sample a random number smaller than $\exp(-(\Delta cost)/\tau)$. $\tau$ controls the temperature and is progressively cooled. We run this simulated annealing procedure for 1000000 iterations with a cooling rate of 0.999995.

**Restriction to the words appearing in librispeech**    In this work, we extensively compare audio and text models. Consequently, we decided to restrict our evaluation to words that appeared at least once in the Librispeech dataset. In the case of synsets associated with multiple words, we preserved only synsets for which all words appeared in the Librispeech dataset. The final dataset is consequently taken from the output of the cross-validation procedure and filtered to preserve only nouns appearing in the Librispeech dataset.

This procedure provided a reasonably large sample of (synset,lemma) pairs: 35,352, while avoiding synsets with major polysemic issues.

EMERGENCE OF SEMANTIC CATEGORY DOES NOT REPLICATE FINDINGS IN TOY MODELS

The sequential emergence of increasingly complex semantic abilities is a property that was initially predicted in toy models Saxe et al. (2014; 2019). In essence, this toy model predicts that larger semantic classes should emerge before precise semantic classes. We adapted these semantic classes by defining semantic categories in the Wordnet hierarchy (Fig A in appendix). To confirm whether these findings extended to transformer models, we then measured whether semantic categories emerged in the order of their size. Surprisingly, we found that this was not the case: class size did not predict the time of emergence of the lexical class (Fig A in appendix). For example, the separation of the main mammal subclasses, like "primate", "marsupial", "rodent", "carnivore", did not precede the separation of carnivore subclasses like "dog" or "feline". On the other hand, this supplementary analysis showed that semantic classes indeed exist, and improve with model size (Fig A in appendix), which strengthens our conclusions on the existence of a semantic graph in language models.

ENGLISH EXPOSED MODEL INSTANTIATE PARTIAL SEMANTIC BETTER THAN CONTROL MODELS

To demonstrate that an English exposed model instantiate partial semantics in a significantly better way than a control model, we measured how well they could classify each semantic category. Remarkably, across most (167, 89%) large semantic categories, the English-exposed model had a better categorization score than control models. We show a scatter plot comparing these scores for early (5) and late (10) layers (Fig C in the appendix panel A-B). This difference increased through the layers of the model, peaking in layer 10 (Fig C in the appendix, panel C). Here, we focus on large categories with more than 100 words to show that this is driven by major semantic categories (a total of 187 categories), but this was also true when we looked at all of the 35532 semantic categories in the dataset.

UMAP VISUALIZATIONS CONFIRM EVALUATIONS OF SEMANTIC SCORES

We plot two UMAP visualisations before and after projection with the 200-D probe (Fig. D in appendix). Before projection, the space is only roughly divided into large semantic categories, presenting many small clusters without a clear organisation. After projection, the space is well organised into continuous clusters mirroring the WordNet tree hierarchy. Consequently, this unsupervised visualisation demonstrates that the WordNet tree is only instantiated in a subspace of the model activity, which is found by our probe.

ADDITIONAL EVALUATION METRICS

We provide in Fig. G additional results of the same syntactic and semantic probe but evaluated with two different metrics from the literature. The Unlabeled Undirected Attachment Score (UAS) measures the percentage of correct syntactic edges recovered by the probe (Hewitt & Manning, 2019). The rank score divides for each node $i$ all other nodes into a positive ($P(i)$) set and negative ($N(i)$) set, depending on whether they are connected to $i$ or not in the graph (semantic or syntactic). The rank score of this node is then the average rank of all positive nodes when sorting all nodes by increasing distance from $i$ (Nickel & Kiela, 2017). The final rank score of a probe is then the median of these ranks scores across all nodes (Nickel & Kiela, 2017).

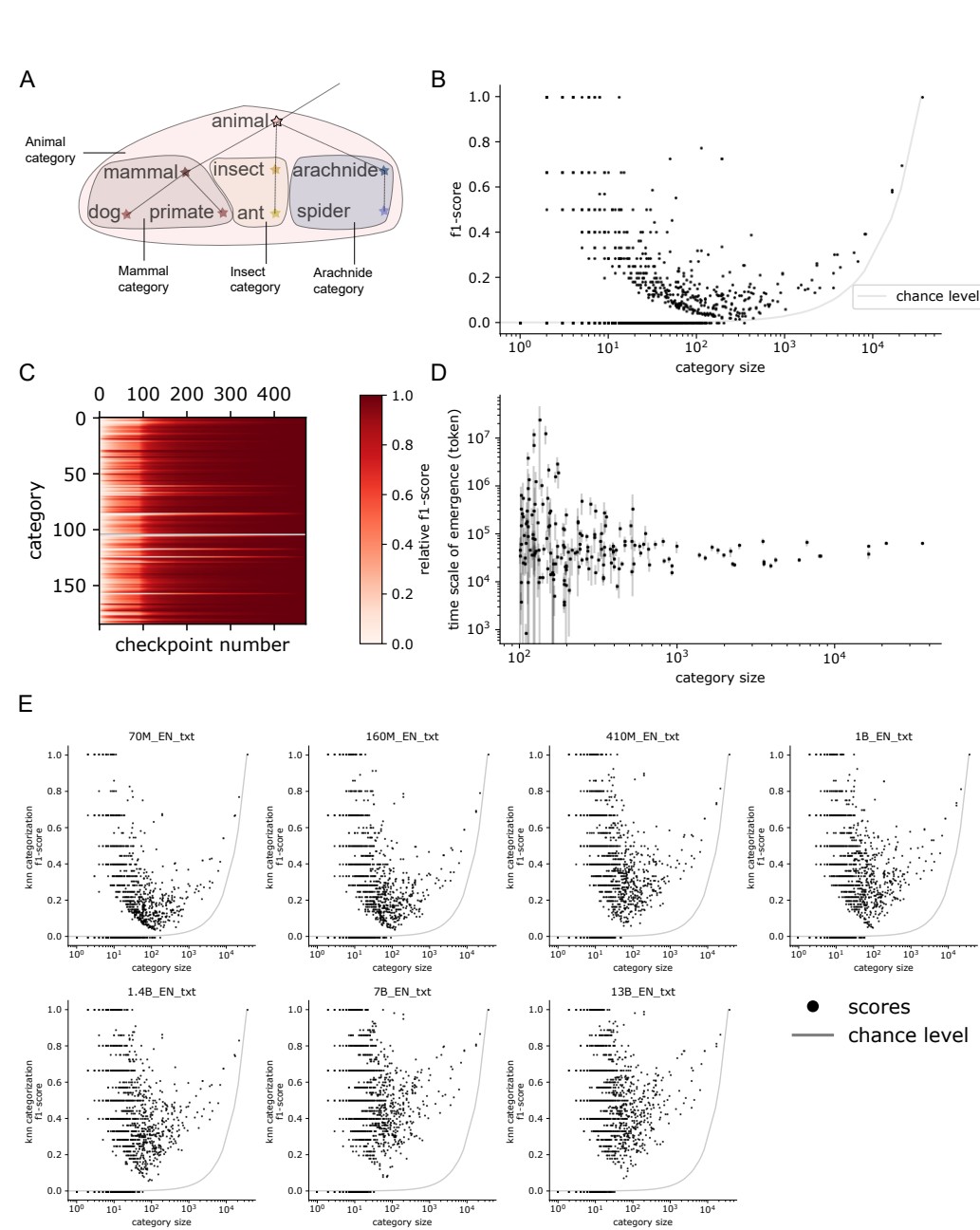

Supplementary Figure A: A: Schematic of the definition of a category in a Wordnet hierarchy. B: f1-score of a 5-Nearest neighbor classification procedure, as a function of category size for a pretrained base Wav2vec 2.0 model. C: f1-score as a function of the checkpoint. D: Time scale of emergence as a function of the category size. Semantic categories do not form in the model in order of their size. E: f1-score for all language models in this study (Pythia suite from 70M to 1.4 billion parameters and Llama 2.0 large language models).

Noun subgraph (n=35352), category >100
Llama 13B, subspace, category prototype

Gold, wordnet subgraph

Supplementary Figure B: Left: 2D projection of the noun subgraph probed from a Llama 2.0 models. Right: gold WordNet noun subgraph.

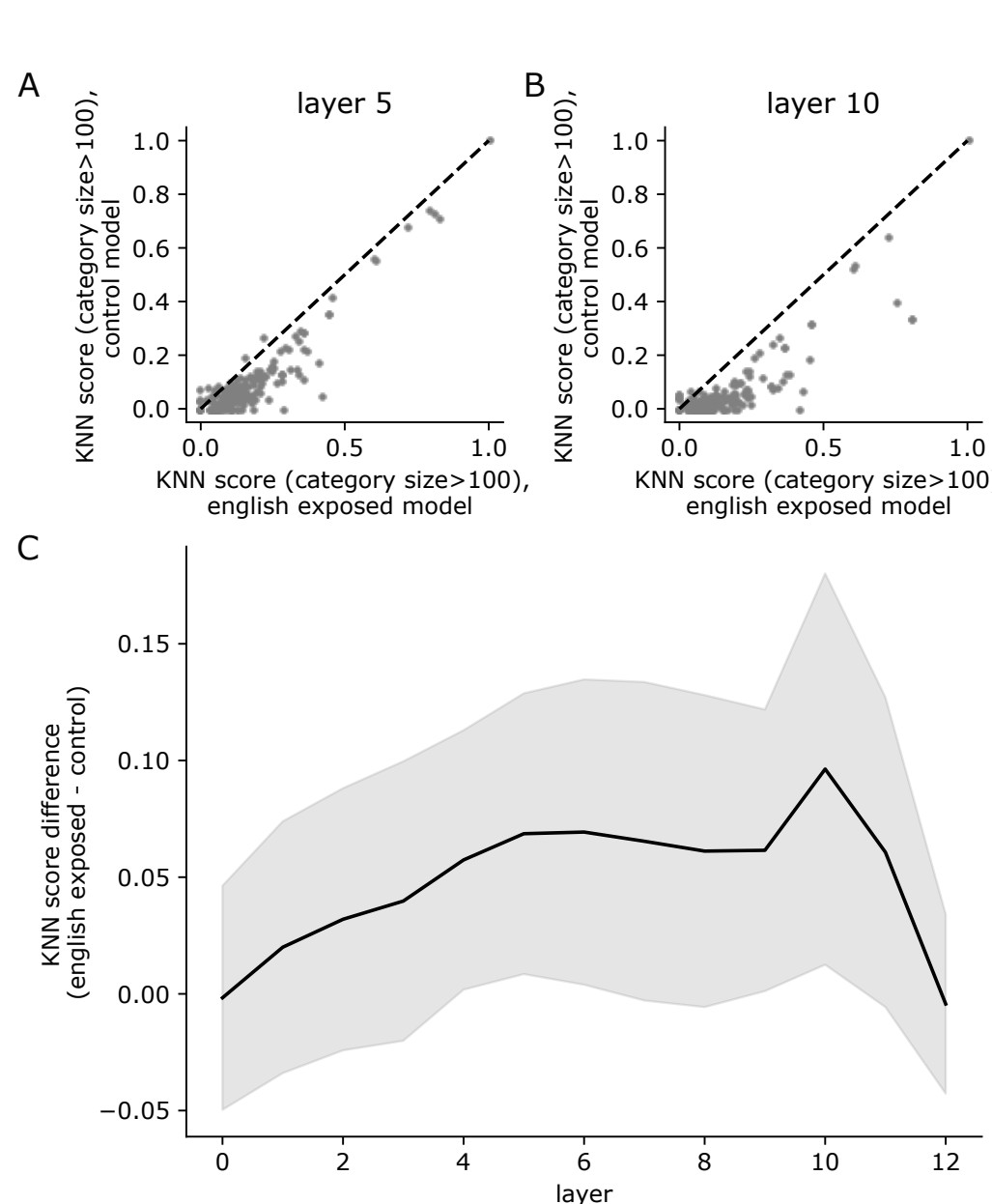

Supplementary Figure C: A-B: Scatter plot of the f1-score of large semantic categories (more than 100 synsets) for an English-exposed Wav2vec 2.0 base model against a control model (Wav2vec 2.0 trained on environmental sounds) for layer 5 (A) and layer 10 (B). C: Difference of f1-score of the KNN classification between the English-exposed and control model, averaged across categories (black line) and with standard deviation (grey shadow) as a function of the layer.

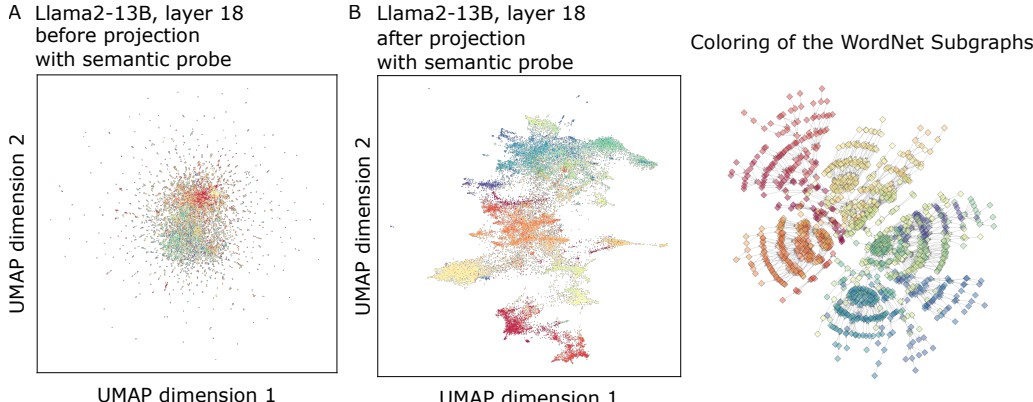

Supplementary Figure D: A: UMAP visualization of the noun subgraph from the activity of a Llama 2.0 model. B: UMAP visualization after projection of a Llama 2.0 model with a 200-D semantic probe.

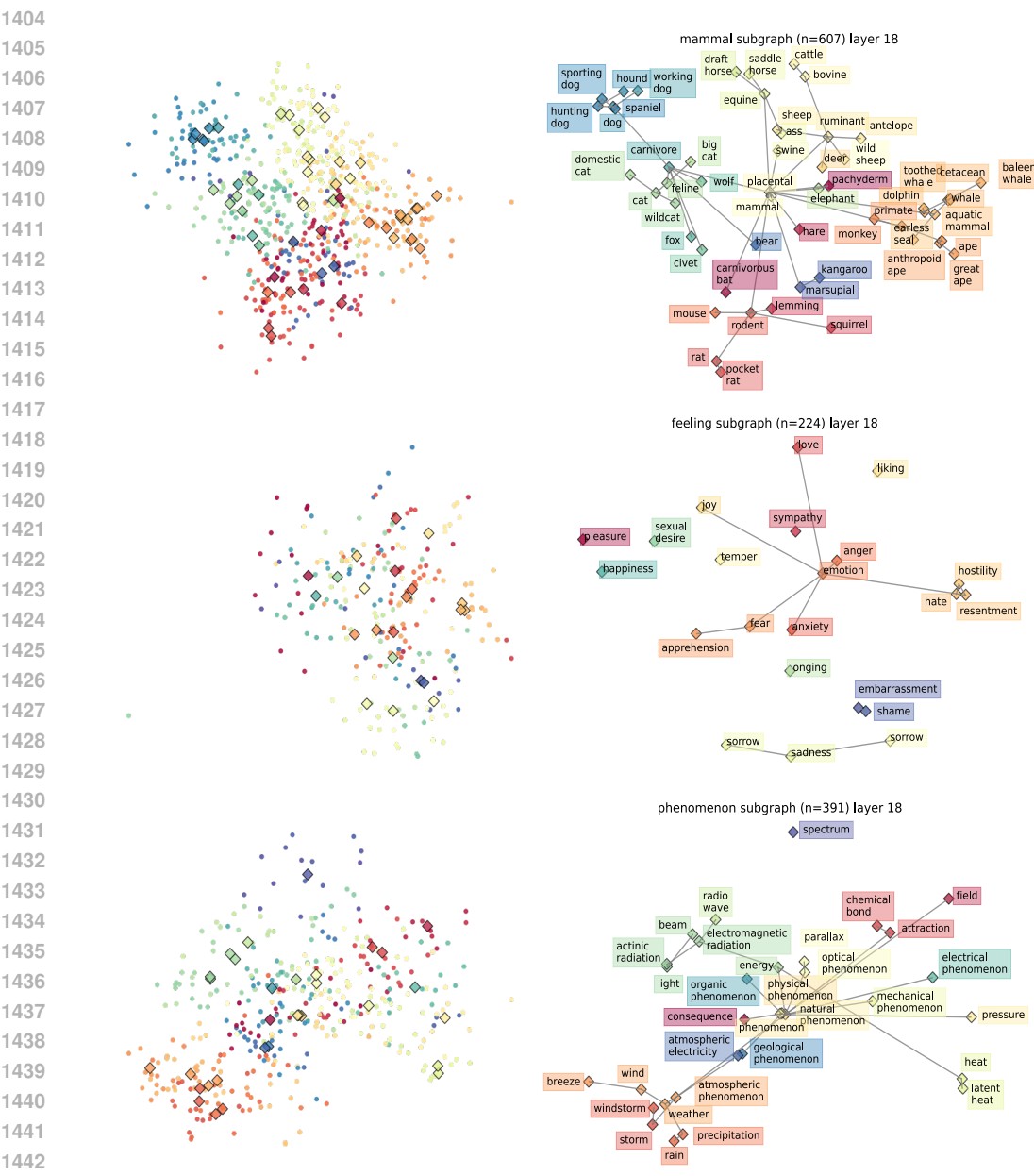

Supplementary Figure E: Llama_13B 2D probing of mammal, feeling and phenomenon subgraphs of Wordnet

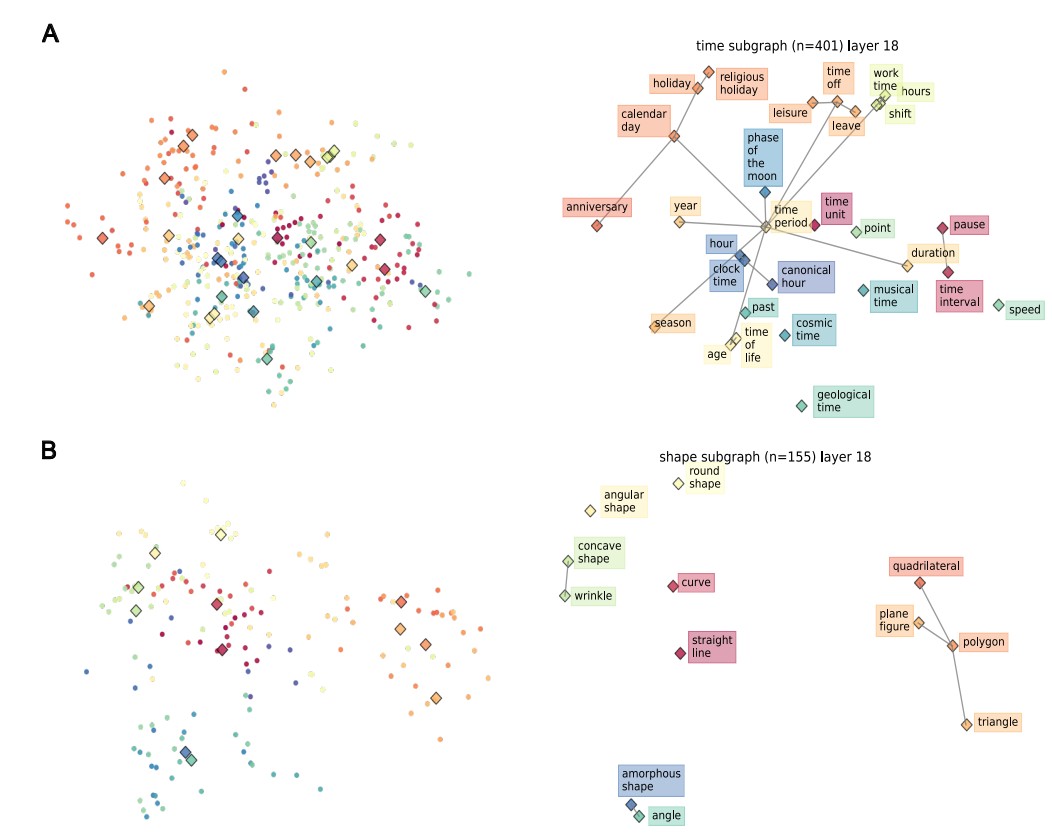

Supplementary Figure F: Llama₋13B 2D probing of time and shape subgraphs of Wordnet

SINGLE UNIT ANALYSES OF SEMANTIC AND SYNTACTIC SUBSPACES

To gain some further insights on how single units interact in the coding of the semantic and syntactic subspace, we additionally ran 3 analyses on the syntactic and semantic probe of a Llama2 model.

We restricted these analyses to model layers for which the semantic and syntactic scores were above 80

First, to understand if two subspaces are aligned, we propose to define an alignment score as the mean square cosines of principal angles between two subspaces (Ake et al, 1973) (i,j): Second, we studied the norm of each unit's probe weights. Third, we quantified how well each unit's activity could be predicted by the semantic and syntactic subspace activity.

We first wanted to quantify whether semantic and syntactic subspaces recruited a similar pattern of units in the model. To do so, we computed the alignment between the left singular vector of each probe. More precisely, we first compute the left singular vector of each probe (over the model units). These vectors quantify how much each unit contributes to the activity of a particular subspace direction. Because they form an orthonormal basis from the same space, we can then compute an alignment score (Ake et al, 1973) as:

$$Alignment(B_i, B_j) = \frac{1}{N}||_i^T V_j||_F^2 = \frac{1}{N}\sum_{n=1}^{N} X_n = \frac{1}{N}\sum_{n=1}^{N} \cos^2(\theta_n) \qquad (1)$$

The alignment score is 0 for orthogonal subspaces (all angles are 90°) and 1 for perfectly aligned subspaces (all angles are 0°). A rough interpretation of in-between scores can be made by taking their arc-cosine. Overall, scores below 0.6 are likely to indicate that the two spaces are not aligned.

We observed a small alignment between the semantic and syntactic subspace of Llama models (max: 0.042, min:0.038)(Fig. H, in appendix, panel A), with a relatively constant value across layers.

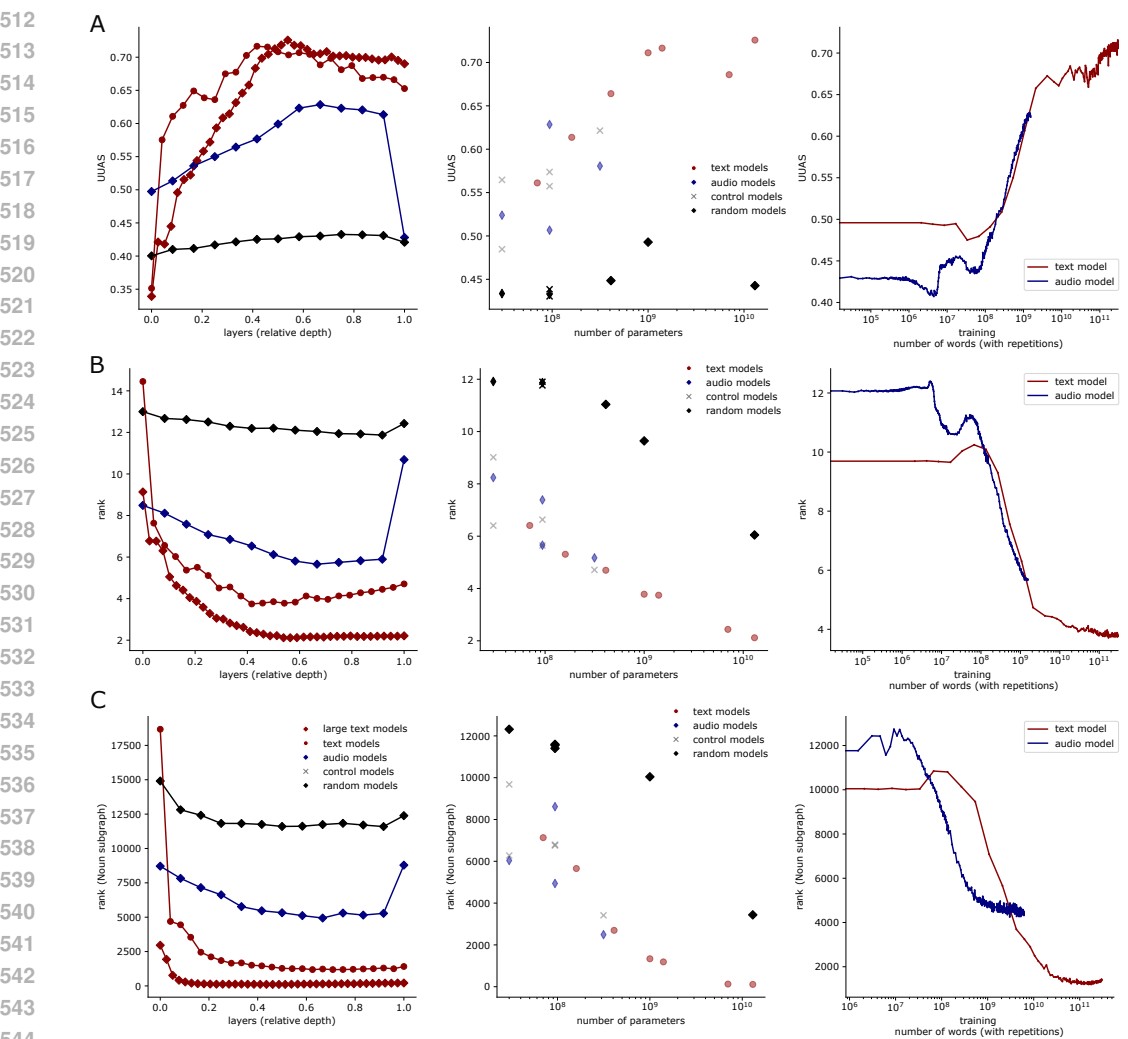

Supplementary Figure G: A: Same as Fig.4, but evaluating the Unlabeled Undirected Attachment Score (UUAS) B. Same as Fig.4, but evaluating the rank score. C: Same as Fig.3, but evaluating the rank score

These results indicate that semantic and syntactic subspaces are mostly unaligned and do not recruit a similar pattern of units in the model.

We next addressed the question of the role of each network unit. We measured for each model unit the norm of its probe vector. (Fig. H, in appendix, panel B) After plotting the distribution of these norms, we observed that probes weights were spread across all units of the layer. This is expected as, following previous work (Hewitt & Manning, 2019), we do not explicitly regularize the norm of the probe, which could have favored the discovery of a sparser solutio We also observed more outliers for the syntactic subspace than the semantic subspace. Some units (40 to 90, depending on the layer) had outlier norm, ranging from 2 to 3 times the median norm. Some units (max 20) were both outliers for the syntactic and semantic subspace. We then repeated these measurements across syntactic and semantic-coding layers of the model (Fig. H, in appendix, panel C), which revealed that an increasing number of units were both outliers for the syntactic and semantic subspace.

These results suggest that the representation of syntactic and semantic subspace is widely distributed across the network units, with a few sets of units being more strongly read out by the probe, although their contribution is mostly washed out by the influence of the majority. Consequently, these sub-

spaces form distributed and distinct, although low-dimensional, patterns of activity in the initial activity space.

To better understand the activity of each individual units, and how we could understand them in terms of syntactic and semantic subspace, we then turned to an encoding approach Indeed, the previous analysis on the optimized probe is not entirely convincing, as the probe is not regularized, such that it could give a high norm to two units while making their contribution cancel out. To deal with that issue, we tried to predict the model unit response from the activity of the syntactic and semantic subspace. Indeed, if some units' activity can be predicted and explained by linear combination of the subspace activity, but not by simpler features, they are likely strongly participating in the syntactic or semantic code.

To remove contamination due to training and testing on particular datasets, we explore a novel set of sentences taken from the Podcast ECOG dataset Zada et al. (2025). This dataset is essentially a transcription of a 30-minute long podcast composed of 5305 words. We gathered the activations of Llama2 by inputing block of 100 words from the podcast transcript in the model and gathering the average activity of all tokens associated with each individual words. We then used the semantic and syntactic probed previously estimates to project each layer activity into a semantic and syntactic subspace. For this analysis, we only kept the 34 layers out of the 40 layers for which the semantic and syntactic scores were above 80% of their respective maximal scores across layers. This was done to make sure the subspace instantiated true semantic and syntactic structure rather than another projection of the model activity.

We then ran an independent ridge regression with 5 outer and 5 inner cross-validation folds (sequential fold block), from the subspace activity to each individual unit response. We then partitioned the variance of each regression into the variance explained by the semantic and syntactic subspace $R2 = R2_{semantic} + R2_{syntax}$. Remarkably, a small number of units (36 in layer 15) had variance strongly predicted by the activity of the syntactic subspace ($R2_{syntax} > 60\%$). Interestingly, these units matched with the outlier units found in the previous analysis (Fig. H, in appendix, panel D).

We then controlled this regression for the effect of known univariate syntactic features. For that purpose, we used the 96 univariate "syntactic features" available with the Podcast dataset, which are made as one-hot encoding of part-of-speech and dependency relationship categories. These features are supposed to encode syntactic properties while not capturing the distance of the syntactic tree. We ran two regressions, one with only the univariate features as predictors, and the second with the univariate features and the subspace activity as predictors. (Fig. H, in appendix, panel E) Remarkably, adding the subspace activity strongly increased the variance explained, indicating that syntactic features were not sufficient to explain these units' responses. Crucially, this was the case in outlier units, whose variance was consequently explained by the syntactic tree-coding subspace instead of univariate semantic features. We additionally verified by eye that these units had no outlier response, computing their mean response across each part-of-speech or dependency category; no clear explanation of their tuning could be found in this manner. In addition, these units' responses were not strongly correlated across the words of the dataset (mean correlation: 0.002, max across two units: 0.39, standard deviation: 0.125), stressing that they had distinct roles in the encoding of syntactic information. We repeated this analysis across all syntax-coding layers of the model (Fig. H, in appendix, panel F), finding a similar conclusion.

Unlike for syntax, no such effect was observed for the semantic subspace: this subspace of activity was not sufficient by itself to predict a large variance of any single unit response.

Together, these results suggest that the instantiation of semantic structure is a distributed process, spread across the units of the model at each layer. The instantiation of syntactic structure is also a distributed process, but with a few units specializing in syntax-coding.

### PROBING TOPOLOGY INSTEAD OF DISTANCES

To complement our distance-based probe, we designed a novel contrastive probe that optimizes the extraction of the topology of the structure, irrespective of the distance. This probe is inspired by contrastive learning approaches of embedding on WordNet (Nickel & Kiela, 2017), and more generally by contrastive learning objectives used in word-embedding vectors.

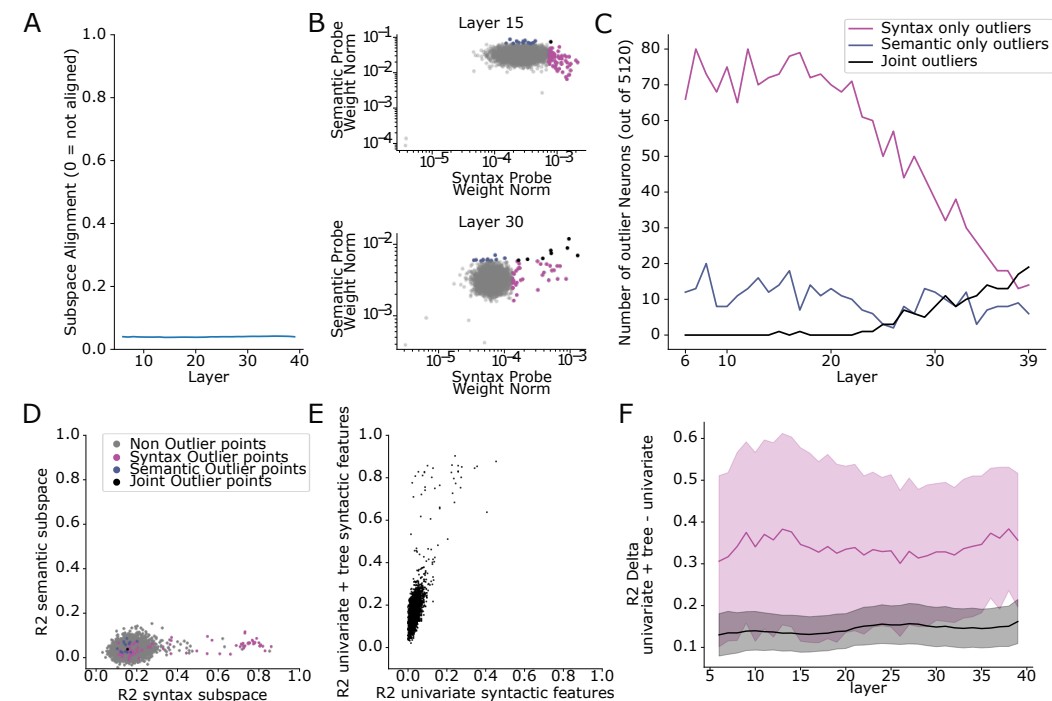

Supplementary Figure H: Subspace alignment and single unit analyses. A: Subspace alignment between the syntactic and semantic subspace across layers instantiating syntax and semantic structure of a Llama2-13B model. B: Scatter plot of syntax and semantic probe norm for all units of layer 15 (top) and layer 30 (bottom) of a Llama2-13B model. Units with outlier probe weight norm are highlighted in purple (syntax), navy (semantic) and black (joint outliers). C: Number of outlier units for syntax, semantics, and joint outliers across layers instantiating syntax and semantic structure of a Llama2-13B model. D: Encoding R2 score of each unit of layer 15 of a Llama2-13B model from the syntactic subspace activity and semantic subspace activity (R2 partitioning). To show that outlier units of the structural probe were indeed explained by the syntactic subspace activity, these units' markers are colored in purple. E: Unique contribution of the syntactic subspace activity compared to simpler univariate syntactic features for each unit of layer 15 of a Llama2-13B model. F: Difference of R2 score (subspace + univariate features - univariate features) across layers instantiating syntax of a Llama2-13B model.

CONTRASTIVE PROBE OBJECTIVE.    The contrastive probe optimizes the contrastive objective of (Nickel & Kiela, 2017), where we replaced the embeddings with a linear projection from the model activity, and optimize the projection. More precisely, given a node $i$ from a graph $S$, we define the set of positive nodes $P(i)$ as the nodes directly connected to this node. We define set of negative nodes $N(i)$ as all other nodes. We then optimize the following objective:

$$\hat{B} = argmin_{B \in \mathbb{R}^{k,p}} \sum_{i \in S} -\log \left( \frac{\sum_{j \in P(i)} \exp(-||(h_i - h_j)B||_2^2)}{\sum_{k \in N(i)} \exp(-||(h_i - h_k)B||_2^2)} \right) \tag{2}$$

In practice, to compute this objective efficiently, we use batched gradient descent, sampling a positive node and a set of negative node. All hyperparameters searches are performed identically to the distance-based probe, with the exception of the number of epochs, which was the same for syntactic probes, but increased to 1000 epochs for semantic probes. Indeed, we observed a slower convergence of the semantic probe with the contrastive objective in term of the number of epochs, but since the computational cost of each epoch is low, we could afford this increase in the number of optimization epochs. The contrastive probe is indeed more memory efficient than the distance-based probe, as it does not require to compute and store all pairwise distances between nodes.

EVALUATIONS    Unlike distance-based probe, the contrastive probe does not optimize the distances between all pairs of nodes. Consequently, it is expected that distance-based evaluations, like spearman correlation show lower scores for this probe. Instead, it should reach higher score on topological evaluations, we use the Unlabeled Undirected Attachment Score (UUAS) for syntax, and rank-based evaluations for syntax and semantics.

RESULTS    Results obtained with the contrastive probe confirmed the results obtained with the distance-based probe. The topology of the semantic WordNet graph emerged in middle-late layers of both text and audio models (Fig. I in appendix, panel A). English-exposed model showed better instantiation of this topology than music-exposed, environmental sounds-exposed or French-exposed control models (Fig. I in appendix, panel B). And the instantiation of this topology increased with model size (Fig. I in appendix, panel B). As observed with the distance-based probe, instantiation of semantic topology progressively emerged during training (Fig. I in appendix, panel C), and saturated for the audio model. Similar results were obtained for the syntactic tree topology (Fig. I in appendix, panels D to I), where we evaluated the instantiation of the syntactic structure through UUAS and rank scores. Distance-based probe and contrastive probe led to similar qualitative conclusions although they offer difference quantitative scores because they optimize different objectives (Fig. J in appendix). The distance-based probe led to higher scores for distance-based evaluations (spearman correlation), while the contrastive probe led to higher scores for topological evaluations (UUAS and rank scores).

Together these results confirm the validity of our measurements using the distance-based probe.

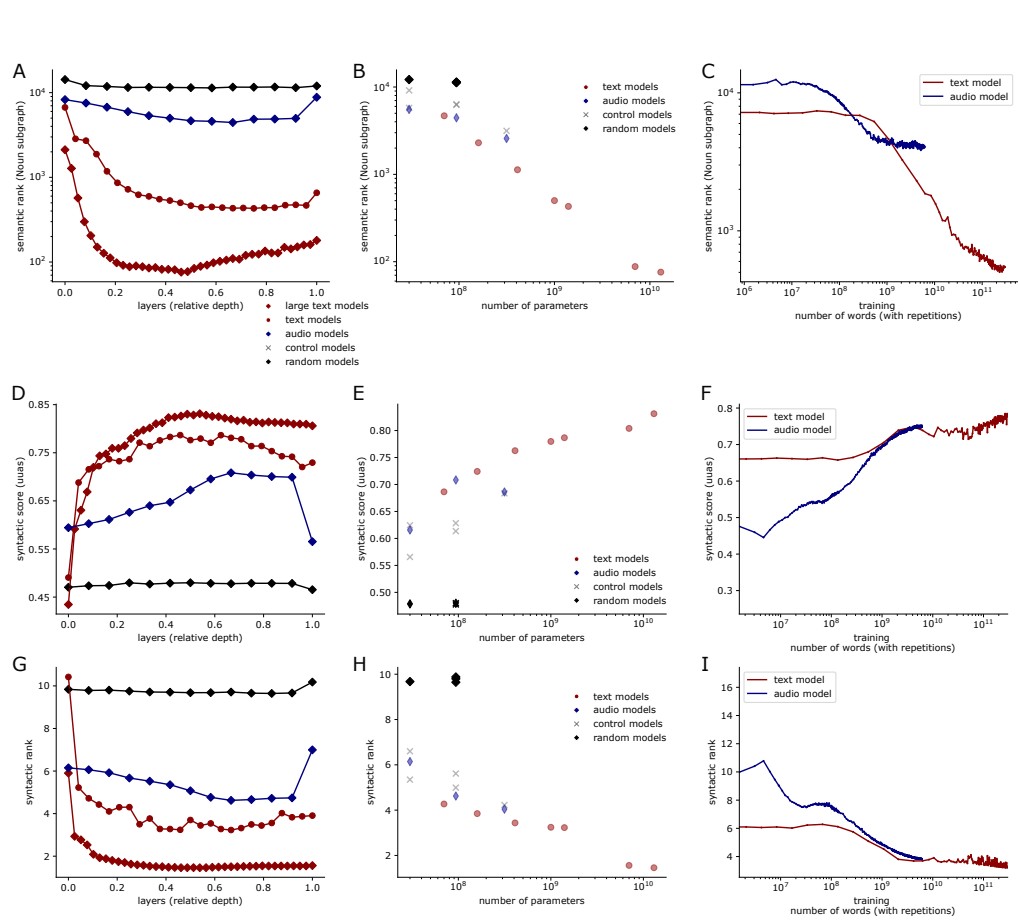

Supplementary Figure I: A: Semantic rank score of a contrastive probe for a large text (Llama-13B), text (pythia-1.4B), audio (Wav2vec 2.0-94M base), and random (Wav2Vec 2.0-94M base) models. B: Semantic rank scores of a contrastive probe for all models as a function of the model size. C: Semantic rank score of a contrastive probe as a function of the quantity of pertaining for the text (pythia-1.4B) and audio (Wav2Vec 2.0-94M base) models. D-F: Same as A-C but for the syntactic dataset evaluated with the UUAS score. G-I: Same as D-F but for the rank score.

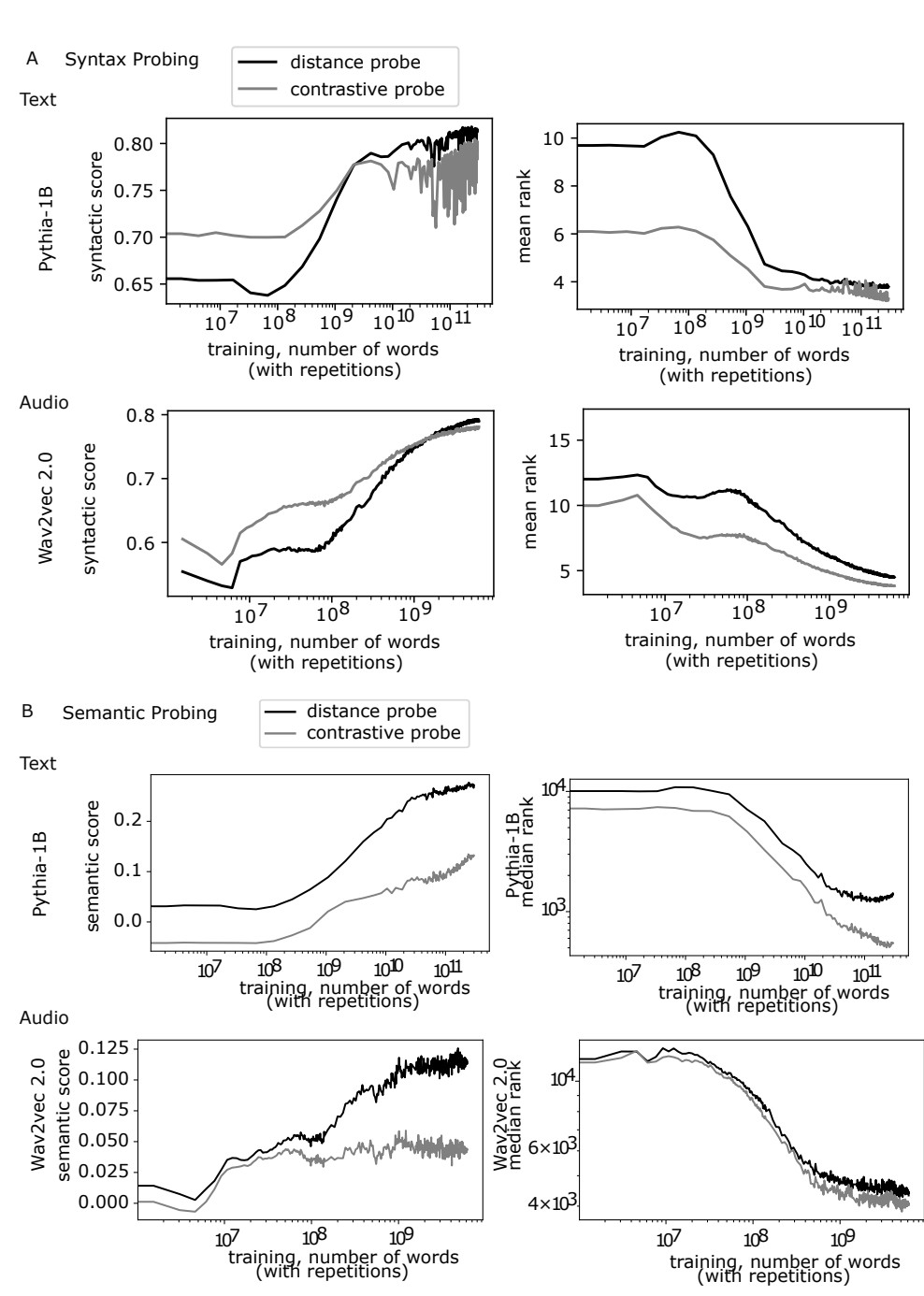

Supplementary Figure J: A: Score of the structural (distance-based) and contrastive probe on the syntax dataset for a text (pythia-1.0B) and audio (Wav2vec 2.0-94M base) model. From left to right, we report for every checkpoint the spearman correlation (syntactic score) and mean rank score. B: Same as A, but for the semantic dataset, reporting spearman correlation (semantic score) and median rank score.

