# OpenReview forum: "Sequential Emergence of Phonemic, Syntactic, and Semantic Representations in Neural Networks"
_ICLR.cc/2026/Conference — Submitted to ICLR 2026_

### Official Review · Reviewer_syiK · 2025-10-31

**Soundness:** 4
**Presentation:** 4
**Contribution:** 4
**Rating:** 10
**Confidence:** 4

**Summary:**

This paper studies the emergence of phonemic, lexical, and syntactic structure in LLMs over the course of training, and compare the resulting dynamics to patterns of acquisition in children language learning. It finds that speech and text models exhibit a staged trajectory through these types of knowledge, first learning phonemic, next lexical, and finally syntactic structure. It further shows that while these features qualitatively are similar to children, quantitatively there is a ~3 order of magnitude difference in number of samples required.

**Strengths:**

This paper conducts a very difficult measurement with admirable rigor and clarity.

The resulting message is potentially quite significant, suggesting that LLMs recapitulate qualitative features of language acquisition, and therefore understanding their acquisition dynamics may contribute to understanding human language learning (of course recognising the considerable quantitive difference which remains).

The paper is clearly written and easy to follow, with figures and experiments of a high standard.

**Weaknesses:**

The paper could be strengthened by disaggregating the studied phenomena into subcategories (specific syntactic categories, etc), but this is reasonable to leave for future work.

The distinctions between phonemics, syntax and semantics rely on particular definitions that come from linguistics and word net, etc. It is well-demonstrated that these distinctions are represented in the network. But one could ask whether a different partitioning of inputs would yield an even clearer developmental trajectory with sharper distinctions. That is, to what extent should we trust these categories (particularly the specific syntactic categories posited)? Again this is reasonable to leave for future work.

**Questions:**

How do learning dynamics and layer scores interact? Do all layers learn at the same time or at different times?

The paper links to toy models of hierarchical learning, which can have a coarse to fine dynamic. However, it is known that basic level categories can emerge first, perhaps due to frequency effects ('dog' is seen far more than carnivore). This effect was also modelled in Saxe et al 2019. Is there evidence for this aspect of human language development?

---

> ### Author Response · Authors · 2025-11-21
> **Response to weaknesses**
>
> We would like to thank the reviewer for his detailed comments and interesting suggestions, stressing that the paper “conducts very difficult measurement with admirable rigor and clarity”, is “clearly written”, “with figures and experiments on a high standard”.
>
> **Weakness 1**
> *"The paper could be strengthened by disaggregating the studied phenomena into subcategories (specific syntactic categories, etc), but this is reasonable to leave for future work."*
>
> We agree with the reviewer that disaggregating the studied phenomena into subcategories will be extremely important and will probably bring us closer to a mechanistic understanding of these emergences.
> An important piece of experimental observation comes from Friedmann et al. 2021, where a clear decomposition of syntactic dependency abilities into 3 stages is made.
> Consequently, one prediction is that, if we manage to adapt the probing approach to dependency trees, we should measure a 3-stage emergence in the model behaviour and representations, unless the model learn differently from children.
> This is definitely some future work that we are planning!
>
> *Reference: Friedmann, N., Belletti, A., & Rizzi, L. (2021). Growing trees : The acquisition of the left periphery. Glossa: A Journal of General Linguistics, 6(1).*
>
> **Weakness 2**
> *"The distinctions between phonemics, syntax and semantics rely on particular definitions that come from linguistics and word net, etc. It is well-demonstrated that these distinctions are represented in the network. But one could ask whether a different partitioning of inputs would yield an even clearer developmental trajectory with sharper distinctions. That is, to what extent should we trust these categories (particularly the specific syntactic categories posited)? Again this is reasonable to leave for future work."*
>
> This is an important point that merits an investigation in future work. If we are to blindly generalize theoretical work from Saxe et al. 2014 in a linear model, the best categories could be obtained by looking at the singular vectors of the representations at convergence.
> We had initially tried some exploration in that flavor, but they never gave satisfactory results, and consequently, we switched back to well-defined linguistic structures with an expected order: phoneme followed by lexical-syntactic emergence.
> As proposed by the reviewer, a different (although related) partitioning is found by Orhan et al. 2025. The author measured that the emergence time of an auditory algebraic structure is correlated to the “complexity” (how strongly it can be compressed) of that structure.
> Despite these two experimental pieces of evidence that some partitionings are well correlated with the emergence time, the community has yet to find a principled approach to identify the partitioning that would be the most predictive of the emergence dynamic.
>
> *Reference:  Orhan, P., Boubenec, Y., & King, J.-R. (2025). The detection of algebraic auditory structures emerges with self-supervised learning. PLOS Computational Biology, 21(9), e1013271.;
> Saxe, A. M., McClelland, J. L., & Ganguli, S. (2014). Exact solutions to the nonlinear dynamics of learning in deep linear neural networks. arXiv:1312.6120*
>
> (Responses to questions are given in a separate comment.)

---

> ### Author Response · Authors · 2025-11-21
> **Response to questions**
>
> **Question 1**
> *"How do learning dynamics and layer scores interact? Do all layers learn at the same time or at different times?"*
>
> We measured this layer-wise dynamic for a Wav2vec 2.0 model before running a hyperparameter search for the probe optimization parameters, so the following results are to be taken with a grain of salt.
> We observed that at initialisation and up to $2\times10^8$ tokens, the early layers are better at instantiating the syntax tree than later layers.
> Between $2\times10^8$ and $10^9$ tokens, this hierarchy is progressively reversed, with the final layers progressively overtaking the initial layers.
> During that time, all layers' syntactic scores grow from 0.5 to 0.75, such that the overall dynamic across layers is roughly the same.
> Finally, between $10^9$ and the end of optimization, the final hierarchy in the layers stabilizes with layers 8-9 topping the hierarchy, while the scores of the early layer and of the final layer 12 remain quite stable.
>
> As a conclusion, all layers learn at roughly the same time, with a slight ordering across layers:
> early layers grow and stabilize in the first half of the training, while mid to late layers keep improving in the second half of the training.
>
> **Question 2**
> *"The paper links to toy models of hierarchical learning, which can have a coarse to fine dynamic. However, it is known that basic level categories can emerge first, perhaps due to frequency effects ('dog' is seen far more than carnivore). This effect was also modelled in Saxe et al 2019. Is there evidence for this aspect of human language development?"*
>
> This is an important effect indeed. Similar to the category size, we correlated the frequency of the word associated with each category with the emergence time of this category, but found no correlation. Nevertheless, this is perhaps due to our definition of a category, which involves classifying a set of words, including frequent and infrequent ones.
> This issue also highlights that our definition of lexical semantics is restrictive: if dog appears very often, it is expected that some understanding of what a dog is, including the features of a dog (four legs, that barks …), should be perhaps learned before the understanding of what a carnivore is.
> Future work should perhaps question more deeply the relationship between such features and semantic graphs like WordNet, and how both can be assessed in a unified way in the representations.

---

### Official Review · Reviewer_VF6d · 2025-11-01

**Soundness:** 2
**Presentation:** 3
**Contribution:** 2
**Rating:** 4
**Confidence:** 3

**Summary:**

This work analyzes text and audio language models in terms of the emergence of phonetic, syntactic and semantic representations using structural probes. Basic idea of the structural probe is to train a small model to quantify the distance of two elements, e.g., syntactic distance of two tokens quantified by the number of edges, and evaluate the prediction by the gold distance created by linguists.
Analyses show that:

- Phonetic distance measured by the difference of articulation features is encoded in the audio models in earlier training stage.
- Semantic distance measured by the number of edges in WordNet is also encoded in text models, but not in audio models.
- Syntactic distance measured by the number of edges in Universal Dependency is encoded in both models.

In addition, the more training and the more parameters imply the better capacity for those linguistic features, and such emergence appear in the earlier stage of training.

**Strengths:**

- Interesting analysis of text models, i.e., LLama2 and Pythia, and audio models, i.e., Wav2Vec, in terms of the linguistic features, i.e., phonetic, lexical semantics and syntactic features. The findings sound very natural to me.
- Analyses are carried out systematically using carefully designed linguistic probes of employing distances defined over phonemes, syntax, lexical semantics.

**Weaknesses:**

- No phoneme level analysis is performed by text models, e.g., Pythia. Given that the analyses for the lexical semantics and syntax are carried out on audio models, it is not clear why the similar analysis is not performed for text models.
- Analyses are performed for three features, syntax, lexical semantics and phoneme, but their relations are not clear, i.e., whether they are correlated or not, although there exist some studies in Figure 5. For example, it is possible to analyze the relationship in terms of Pareto optimality [1]. It is also interesting to figure out what neurons are primarily triggered for particular features by analyzing the learned matrices, B, to find the relations of those linguistic features.

[1] Jiannan Xiang, Huayang Li, Defu Lian, Guoping Huang, Taro Watanabe, Lemao Liu. Visualizing the Relationship Between Encoded Linguistic Information and Task Performance. EMNLP 2022 Findings.

**Questions:**

- Some details are missing:
  * Phonemic representations are immediately clear in section 2.2, but is it using the IPA sequences given the use of Montreal Forced Aligner?
  * What is meant by 2-dimensional or 2D probes mentioned elsewhere? Do they mean the dimension of B, i.e., p?

---

> ### Author Response · Authors · 2025-11-20
> **Response to weaknesses**
>
> We would like to thank the reviewer for stressing “natural findings” through “interesting analysis”, which were “carried out systematically using carefully designed linguistic probes”.
>
> **Weakness 1**
> *"No phoneme level analysis is performed by text models, e.g., Pythia. Given that the analyses for the lexical semantics and syntax are carried out on audio models, it is not clear why the similar analysis is not performed for text models.*"
>
> The reviewer is right that ideally, our evaluation would cover symmetrically audio and text models.
> Nevertheless, large language models used in this study do not work at the level of phonemes or characters, instead using tokens that are sometimes words, and evaluating them on phonemic representations would not be valid. Additionally, even letter-based text models do not have access to acoustic and, consequently, articulatory features, which means that we do not expect them to develop a structured representation of phonemes according to articulatory features.
>
> We add the following sentences in the limitations paragraph of the discussion:
>
> &nbsp;&nbsp; L500 “Fifth, we do not probe phonemic structure in text models. This is because text models for which checkpoints were available used token-level inputs instead of letters or grapheme inputs.”
>
> In addition to text, extending these analyses to multimodal (speech + reading) models is an exciting direction for future work.
>
> **Weakness 2**
> *"Analyses are performed for three features, syntax, lexical semantics and phoneme, but their relations are not clear, i.e., whether they are correlated or not, although there exist some studies in Figure 5."*
>
> The reviewer is right in highlighting that we have not thoroughly investigated the relationship between phonemic, syntactic, and semantic subspaces elicited by our probes.
>
> We expect that these linear subspaces are largely disjoint.
> To demonstrate this, we computed the alignment between subspaces and reported them in Fig. H, in appendix. We observed a small alignment between the semantic and syntactic subspaces of Llama2 models (max: 0.042 , min:0.038 )(Fig. H, in appendix, panel A), with a relatively constant value across layers. These results indicate that semantic and syntactic subspaces are mostly disjoint and do not share a representational basis.
>
> We add Fig. H and a detailed method L1490-1553 in the appendix to support these claims.
>
> **Weakness 3**
> *"For example, it is possible to analyze the relationship in terms of Pareto optimality [1]"*
>
> We thank the reviewer for pointing out the work of Xiang et al. 2022.
> These findings are valuable because they study how probing and model training can mutually benefit each other to achieve better-performing and more interpretable models. However, throughout our study, we aim to understand the learning dynamics of self-supervised models during pre-training. To do so, we can’t change the parameter of the model, which prevents us from looking at this Pareto-frontier.
>
> We believe adopting such pretraining methodology could help bridge the gap we observe in this study; the emergence of linguistic structures in both text and speech models is too slow and data-hungry.
>
> We add this reference to the sentence:
>
> &nbsp;&nbsp; L431 “This discrepancy highlights that modern language algorithms remain remarkably inefficient, and thus calls for exploring novel neural architectures and training paradigms”
>
> *Reference: Jiannan Xiang, Huayang Li, Defu Lian, Guoping Huang, Taro Watanabe, Lemao Liu. Visualizing the Relationship Between Encoded Linguistic Information and Task Performance. EMNLP 2022 Findings.*
>
> **Weakness 4**
> *"It is also interesting to figure out what neurons are primarily triggered for particular features by analyzing the learned matrices, B, to find the relations of those linguistic features."*
>
> As suggested by the reviewer, we next addressed the question of the role of each network unit. We measured for each model unit $ i$ the norm of its probe vector $B[i,:]$. (Fig. H, in appendix, panel B-C)
> This analysis revealed that some units (10 to 70) have outlier norms, and more so for the syntactic probes than for the semantic ones.
>
> We confirmed that these outlier neurons are dedicated to structure processing with a regression analysis. We attempted to reconstruct the activations of the model units given the subspace activity. The syntactic subspace was sufficient to predict about 80% of the syntax outlier units’ activations. Semantic outliers, on the other hand, were poorly predicted by the semantic subspace.
>
> Together, these results suggest that the instantiation of semantic structure is a distributed process, spread across the units of the model at each layer. The instantiation of syntactic structure is also a distributed process, but with a few units specialising in syntax-coding.
>
> We add Fig. H and methods L1554 to L1611 in the appendix to support these claims.

---

> > ### Comment · Reviewer_VF6d · 2025-11-27
> >
> > Thank you for additional inputs.
> >
> > > large language models used in this study do not work at the level of phonemes or characters, instead using tokens that are sometimes words, and evaluating them on phonemic representations would not be valid.
> >
> > Given the training data from Pythia comes from Pile, it should already have IPA symbols. Also, given the alignment between text and IPA sequences, it would be possible to probe text inputs for phoneme level using the alignment from text to IPA. Or, directly feed IPA sequences to see any alignment between phonemes.

---

> > > ### Author Response · Authors · 2025-11-28
> > > **Evaluation of text models on IPA sequences**
> > >
> > > The reviewer is right that IPA symbols have been seen by the Pythia models during their training.
> > > The tokenizer of the models encoded each symbol with 1 (21 IPA symbols), 2 (18 IPA symbols), 3 (8 IPA symbols), or 4 (2 IPA symbols) tokens. The average of each of these sets of tokens was a unique embedding vector, such that the Pythia models perfectly separated phonemes already at the stage of tokenization (1NN-classification of IPA symbols score: 1.0).
> > >
> > > Following the request of the reviewer, we converted our phoneme dataset (UD–EWT–TTS) into IPA sequences and probed phonemic structure in trained and untrained Pythia models.
> > > When tested on left-out phonemes from the test set, randomly initialized and trained Pythia models reached a similar score, as reported in the table below (max Spearman correlation across layers). These results confirm our hypothesis that Pythia models do not instantiate a phonemic structure, despite perfectly separating IPA symbols because of their distinct tokenization.
> > >
> > >    | Model      | untrained phonemic score  | trained phonemic score |
> > >    |--------------|-------------|-------------|
> > >    | Pythia-160M     |     0.43  |      0.43  |
> > >    | Pythia-410M  |     0.50  |     0.51  |
> > >    | Pythia-1B |      0.40  |     0.41  |
> > >    | Pythia-1.4B |      0.47  |     0.48  |

---

> ### Author Response · Authors · 2025-11-20
> **Response to questions**
>
> **Question 1**
> *"Phonemic representations are immediately clear in section 2.2, but is it using the IPA sequences given the use of Montreal Forced Aligner?"*
>
> Indeed, as pointed out by the reviewer, Montreal Forced Aligner returns phonemes in ARPABET format. We converted phoneme symbols from ARPABET to IPA using the panphon Python package.
>
> We add the following sentence to the main text to make this cleaner:
>
> &nbsp;&nbsp; L156 “Phoneme symbols were converted from ARPABET to IPA using the panphon library”
>
> *Reference: David R. Mortensen, Patrick Littell, Akash Bharadwaj, Kartik Goyal, Chris Dyer, and Lori Levin. PanPhon: A resource for mapping IPA segments to articulatory feature vectors. Proceedings of COLING 2016, the 26th International Conference on Computational Linguistics: Technical Papers, pp. 3475–3484*
>
> **Question 2**
> *"What is meant by 2-dimensional or 2D probes mentioned elsewhere? Do they mean the dimension of B, i.e., p?"*
>
> A 2-dimensional probe is a probe that projects the activity of the model into a space of dimension 2; the probe is consequently a matrix $B$ of size $(k,2)$  where k is the dimension of the activation vectors.
> To make this clearer, we add the following statement in the method:
>
> &nbsp;&nbsp; L114 “We optimize 2D ($B\in \mathbb{R}^{k,2}$) probe for visualization and 200D ($B\in\mathbb{R}^{k,200}$) probe for evaluation.”

---

### Official Review · Reviewer_RCBG · 2025-11-01

**Soundness:** 2
**Presentation:** 1
**Contribution:** 2
**Rating:** 2
**Confidence:** 4

**Summary:**

This paper has three main aims according to my reading: (1) applying the structural probe of Hewitt and Manning to linguistic structures outside of syntactic dependency parse trees, specifically to distances between English vowels based on their articulatory features and distances between English nouns based on WordNet graph; (2) analyzing whether these distances (including syntactic distance) are represented in language models trained on text or speech; and (3) if they are, what their developmental trajectories look like.

The main findings as claimed by the paper are as follows. Signatures of three levels of linguistic representation (phonemic, syntactic, lexical semantic) can be found in text and speech models. The signal strength for lexical semantic representation is weaker in speech models, and the signal present seems to derive mostly from phonemic subsequence overlap. When analyzing the training trajectory of the speech models, the transitional phase of metric score increases occurs in the order of phonemic - lexical semantic - syntactic scores.

There are definitely interesting sets of raw results in this work. But I think the framing and interpretation of the results are questionable, and sometimes even misleading, as I will discuss in the individual sections.

**Strengths:**

There are definitely new empirical results that people in computational linguistics and cognitive science would be interested in. The research question itself is interesting, and so is looking at the learning trajectories (especially of speech models, which is done less frequently). The implementation of the experiments seemed overall reasonable.

**Weaknesses:**

Overall, how the experiments were set up seemed fair to me as mentioned in the Strengths section (the quantitative findings reported are probably real findings), minus a few questionable parts that I will comment on below. However, the major soundness issue I see in this work is how the experimental results are interpreted and used to support the core claim made. The presentation and exposition also seemed weak, which I will also discuss below separately.

**Soundness**
- Specifically, I take issue with the interpretation that audio models "partially instantiate" lexical semantic structure - to me, to partially instantiate it, it must learn something interesting about it. The numeric difference between control and audio models as shown in Figure 3D seems to be quite small, and the paper itself seems to point out that audio models don't actually seem to be learning anything meaningful about lexical semantics other than the fact that certain labels for semantic categories share phonological subsequences (citing from the paper, the scores of the control models "were in the same range as the semantic scores of models exposed to English speech"). Then, I'm not sure how the current results can be interpreted as "partial instantiation" (L311: "Together, these results demonstrate a clear and partial instantiation of lexical semantic structure by text and audio models, respectively.") unless there were some qualitative evidence and statistical tests showing that there are interesting gains over control models. If there is nothing much substantive here, I don't know how the whole analysis and interpretation in Section 3.4 ("Order of Acquisition"), and very broad claims like "Linguistic structures are acquired in sequential order" are justifiable at all.
- I think the trajectory result still can have a sensible interpretation; maybe having some grasp of phonology is a precondition to being able to realize that there are shared phonological subsequences between expressions. But this is certainly different from saying that lexical semantics emerges in a network after phonology and before syntax, which is one of the core claims of the paper (it's in the title!). I acknowledge that some flavor of this comment exists already in draft too (L420-), but my opinion is that even with this hedging, the current framing is an overinterpretation. And even more problematic to make this interpretation the title of the paper.
- In general, the framing connecting the results to child language acquisition also seems to be misleading. For instance, if we take the paper's words that "the developmental ordering between syntax and lexical semantics, a central question in cognitive science (Gleitman, 1990; Pinker, 1994), remains unresolved.", then how can one justify statements like "Like children, audio models appear to spontaneously build linguistic features, and this emergence follows a stereotypical sequence of phonemic, lexical and syntactic representations."? I think there are actual stakes in making uncareful analogies between human and LM results because it promotes proliferation of unqualified claims/beliefs like "LLMs learn like human children do" - I could totally see this paper being cited in this way when the actual picture painted by the experiments a lot more nuanced.
- Additionally, I see a few representativeness issues in the selection of the domain and method. These aren't problematic per se (the scope of things that one paper can achieve is limited), but becomes problematic when trying to use the results to claim something much more broad.
	- Representativeness of the selected evaluation domains: "phonemic structure" is only limited to articulatory features of vowels, and "lexical semantic structure" is only limited to nouns.
	- Maybe more major but this perhaps reflects my own bias - representativeness of reducing linguistic structural representations to distance. This obviously is true of the original method of H&M too, but I feel like the framing taken in this work about reducing structure to distance is celebratory, rather than viewing it as a limitation. Taking from the paper: "Each level representation can be formalized as a metric system, which indicates e.g. whether different phonemes are more or less similar to one-another or whether different words in a sentence are more or less syntactically related. Formally, this prediction implies that language representations reduce down to a distance matrix." -> Two comments here: (1) is about the perhaps subtle distinction between representation being formalized as distance vs. formalization of linguistic representation in a way that lets you allow distance metrics to be defined between them. I am sympathetic to well thought out ways to achieve the latter but not the former. (2) is about the actual ways in which the latter idea is realized, which can be assessed in terms of their adequacy of capturing linguistic information that we care about. I think the paper's proposed representations and distance metrics miss out a lot, connecting back to representatitiveness issue I mentioned above.

**(Minor) Methodological comment**
- L199: "Only the French large and base models outperform the tiny English model, which is expected given the model’s limited size. These controls prove that the instantiation of phoneme structure results from the emergence of speech-specific processes." I'm not sure if I am convinced. There should be a tiny French model in order to make this claim based on the current results.

**Presentation**

I found the exposition, discussion, and interpretations offered often confusing. Attribution to existing work could also be improved.

Examples of confusing/missing exposition:
- L257: "This hierarchical organization across layers highlights that these semantic embeddings are not present in the audio or text tokens’ embedding used as input to these transformer models. Instead, several computational steps are performed to merge and position these tokens’ embedding together into a semantic subspace." This really needs a lot of unpacking. Why does the (inverse) U-shape indicate "hierarchical organization"? And by semantic embeddings, I'm guessing what is intended is (purported) lexical semantic information? Does the "Instead..." sentence offer anything more than just saying that there is layer-wise processing that contextualizes information with respect to other tokens in the input in a transformer? This is the type of discussion that popped up in many places where the logical flow doesn't sound quite right or the justification for the claim is not quite there.
- L316: "Language models are known to code syntactic trees through distances (Hewitt & Manning, 2019), while only correlates of syntactic code have been found in speech models (Pasad et al., 2024b)." I think I understand what is intended here but I find this kind of phrasing a bit bizarre because to me distances also seem like correlates of syntactic code. Although to be honest I'm not sure my understanding of the term "syntactic code" aligns with how it's used.
- What is the method with which the trees are reconstructed from the distance metric? (e.g., in Fig 4) Minimum spanning tree as in Hewitt and Manning? I'm also curious why UUAS was dropped as a metric for syntactic trees if the tree reconstruction was done.

Examples of missing attribution:
- L206: "This is considerably more than the 10^7 words that children experience in their first year on average." -> What is the source of this claim?
- L214: "Nevertheless, no analysis has shown the existence of a theoretically constructed semantic graph in a subspace of the model activity." -> First of all I'm not sure if I understand what exactly is in the set of "a theoretically constructed semantic graph". But there's plenty of representational studies based on information in semantic graphs and plenty on info from WordNet specifically (although they might not be looking at exactly the formulation of the problem looked at in this paper):
	- https://aclanthology.org/2023.findings-eacl.139
	- https://arxiv.org/abs/2402.16837
	- Maybe more importantly, I think Park et al. (2025) is currently cited after "the activation patterns of self-supervised models represent simple semantic relationships" but their analysis of hierarchical structure of words in WordNet in terms of shortest distance matrix between nouns seem exactly aligned in spirit with the lexical semantics part of this work. Surely there are differences but maybe this analysis at least merits a discussion?

Nitpicky stylistic comments:
- I suggest using proper opening quotation marks in latex, all of them are currently closing quotation marks.
- Figure 2D model identifiers are hard to map onto textual descriptions of the models for a reader who is not familiar with the identifiers. (e.g., I had to backtrace citations to figure out FMA is a music dataset, maybe relabeling them to match the descriptions in the text might help the readers)
- L092: "Each level representation" -> "Each level of representation"
- L118: made of text different text datasets -> made of different text datasets
- L159: "score models with known strong cognitive abilities" seems like an unqualified claim. what does "strong cognitive abilities" mean?
- L204: "Fig. 2D shows a gradual rise" -> 2E?
- L256: "For this 200-D probe, the semantic scores across model layers follow a U-shape in all models" - Do you mean inverse U?

**Questions:**

- If there are any responses to the soundness issues I raised, I'd be happy to engage! The main claim didn't feel fully justifiable to me but I'd be happy to update my beliefs if there is good justification.
- Is it problematic that the phonemic scores for the models trained with nonlinguistic data like music is very high? It being high for French models is understandable, since some vowel distinctions in English will be shared in French vowels, but it feels like a Spearman correlation of >0.5 is almost excessively high for a model trained on music or environmental sounds.
- Given the discussion about linear trees throwing off the metrics on p7 (which I appreciated), maybe it would be good to put the correlation with linear tree directly on Fig 4D as another baseline, eg as a horizontal dotted line on the graph.

---

> ### Author Response · Authors · 2025-11-20
> **Response to soundness issues (1-2)**
>
> We would like to thank the reviewer for pointing out that the results are “interesting” and relevant for people in “computational linguistics and cognitive science” through a “reasonable implementation” of the experiments.
>
> **Soundness 1**
> *"The major soundness issue I see in this work is how the experimental results are interpreted and used to support the core claim made.  Specifically, I take issue with the interpretation that audio models "partially instantiate" lexical semantic structure - to me, to partially instantiate it, it must learn something interesting about it.
> The numeric difference between control and audio models as shown in Figure 3D seems to be quite small, and the paper itself seems to point out that audio models don't actually seem to be learning anything meaningful about lexical semantics other than the fact that certain labels for semantic categories share phonological subsequences (citing from the paper, the scores of the control models "were in the same range as the semantic scores of models exposed to English speech").
> Then, I'm not sure how the current results can be interpreted as "partial instantiation" (L311: "Together, these results demonstrate a clear and partial instantiation of lexical semantic structure by text and audio models, respectively.") unless there were some qualitative evidence and statistical tests showing that there are interesting gains over control models. If there is nothing much substantive here, I don't know how the whole analysis and interpretation in Section 3.4 ("Order of Acquisition"), and very broad claims like "Linguistic structures are acquired in sequential order" are justifiable at all."*
>
> This is a fair point: audio models are notoriously underperformers in semantic abilities compared to text models, although some results show performance beyond acoustic processing (Pasad et al. 2024).
> To address this concern, we now show with statistical tests that English-exposed models significantly surpass control models on a lexical semantic categorisation task:
>
> &nbsp;&nbsp; L300: “Despite their low semantic score, the English-exposed models better clustered 89% of semantic categories of the WordNet tree than control models, with a 0.1 F1-score gain on average (Fig F in appendix,  paired t-test, $p = 3.8\times10^{-36} $ ).”
>
> We add a new paragraph (L1147 to L1157), and Fig. C to the appendix to detail these claims.
> The results confirm a modest but significant instantiation of semantic structure in English-exposed speech models.
>
> *Reference: Pasad, A., Chien, C.-M., Settle, S., & Livescu, K. (2024). What Do Self-Supervised Speech Models Know About Words? Transactions of the Association for Computational Linguistics, 12, 372‑391.*
>
> **Soundness 2**
> *"I think the trajectory result still can have a sensible interpretation; maybe having some grasp of phonology is a precondition to being able to realize that there are shared phonological subsequences between expressions. But this is certainly different from saying that lexical semantics emerges in a network after phonology and before syntax, which is one of the core claims of the paper (it's in the title!). I acknowledge that some flavor of this comment exists already in draft too (L420-), but my opinion is that even with this hedging, the current framing is an overinterpretation. And even more problematic to make this interpretation the title of the paper."*
>
> We did not intend for this paper to suggest that syntax comes before semantics, and indeed, the two summary curves largely overlap. On the contrary, we want to systematically evaluate the amount of data necessary for these representations to emerge, and empirically observe that low-level phonemic structures emerge first.
>
> To avoid any misunderstanding, we updated the discussion paragraph:
>
> &nbsp;&nbsp; 	L422 “This sequential emergence ” → “This emergence ”
>
> &nbsp;&nbsp; 	L470 “This sequential emergence strengthens earlier work on toy models.” → Emergence of simple (phonemic) followed by higher-order (lexical-syntactic) structures strengthens earlier work on toy models.
>
> We updated the “Order of Acquisition section”:
>
> &nbsp;&nbsp;	L376 “ checkpoints that highlight the successive emergence of phonemic, lexical semantic, and
> syntactic structures in the audio models.” → “checkpoints that highlight the successive emergence of phonemic and lexical-syntactic structures in the audio models”
>
> We change the caption of Fig. 5:
>
> &nbsp;&nbsp;	“demonstrating the successive emergence of phonemic, lexical semantic, and syntactic
> structures.” → “demonstrating the emergence of phonemic followed by lexical-syntactic structures.”
>
> We removed ‘sequential” from the title: “Emergence of phonemic, syntactic and semantic representations in artificial neural networks”
>
> (The Response to the third soundness point is made in a separate comment, like the response to representativeness, presentation, and question issues.)

---

> > ### Author Response · Authors · 2025-11-20
> > **Response to soundness issues 3**
> >
> > **Soundness 3**
> > *“In general, the framing connecting the results to child language acquisition also seems to be misleading. For instance, if we take the paper's words that "the developmental ordering between syntax and lexical semantics, a central question in cognitive science (Gleitman, 1990; Pinker, 1994), remains unresolved.", then how can one justify statements like "Like children, audio models appear to spontaneously build linguistic features, and this emergence follows a stereotypical sequence of phonemic, lexical and syntactic representations."? I think there are actual stakes in making uncareful analogies between human and LM results because it promotes proliferation of unqualified claims/beliefs like "LLMs learn like human children do" - I could totally see this paper being cited in this way when the actual picture painted by the experiments a lot more nuanced."*
> >
> > We appreciate this concern, and now revise this statement as:
> >
> > &nbsp;&nbsp; L420: “In spite of their remarkably distinct architecture and learning principles, audio models appear, like children, to spontaneously build linguistic features, in a developmental trajectory that, at least at a coarse view, follows a phonemic and lexico-syntactic sequence.”

---

> ### Author Response · Authors · 2025-11-20
> **Response to representativeness issues**
>
> **Representativeness 1**
> *"Representativeness of the selected evaluation domains: "phonemic structure" is only limited to articulatory features of vowels, and "lexical semantic structure" is only limited to nouns."*
>
> This is a good point.  Regarding part of speech, we originally focused on nouns to avoid a confound. To clarify this issue, we added the following paragraph:
>
> &nbsp;&nbsp; L143: “We focused on the noun subgraph of WordNet because including several parts of speech (POS) categories can lead to a large semantic score just because of an ability to discriminate between POS, and so, mislead to the conclusion that probes have learnt semantic structure.” (e.g. Limisiewicz et al, 2021)
>
> and add the following limitation to our discussion:
>
> &nbsp;&nbsp; L502: “Sixth, our analysis is restricted to nouns to avoid equating semantics and part-of-speech categorization. This analytical choice may limit the generality of the present finding. Future work should thus extend to other parts of speech and explore semantic relationships beyond hypernymy.”
>
> For phoneme structure, we had originally opted to only plot the vowels for clarity.
> We have run the analysis for consonants on one model, which showed a similar qualitative trend, and will provide the results for all consonants as soon as possible during this rebuttal.
>
> *Reference: Limisiewicz, T., & Mareček, D. (2021). Introducing Orthogonal Constraint in Structural Probes. In C. Zong, F. Xia, W. Li, & R. Navigli (Éds.), Proceedings of the 59th Annual Meeting of the Association for Computational Linguistics and the 11th International Joint Conference on Natural Language Processing (Volume 1 : Long Papers) (p. 428‑442). Association for Computational Linguistics.*
>
> **Representativeness 2**
> *"Maybe more major but this perhaps reflects my own bias - representativeness of reducing linguistic structural representations to distance. This obviously is true of the original method of H&M too, but I feel like the framing taken in this work about reducing structure to distance is celebratory, rather than viewing it as a limitation. Taking from the paper: "Each level representation can be formalized as a metric system, which indicates e.g. whether different phonemes are more or less similar to one-another or whether different words in a sentence are more or less syntactically related. Formally, this prediction implies that language representations reduce down to a distance matrix." -> Two comments here: (1) is about the perhaps subtle distinction between representation being formalized as distance vs. formalization of linguistic representation in a way that lets you allow distance metrics to be defined between them. I am sympathetic to well thought out ways to achieve the latter but not the former. (2) is about the actual ways in which the latter idea is realized, which can be assessed in terms of their adequacy of capturing linguistic information that we care about. I think the paper's proposed representations and distance metrics miss out a lot, connecting back to representatitiveness issue I mentioned above."*
>
> We agree with the reviewer that distance codes are not necessarily the best code, neither for semantic nor syntactic structure.
> We consequently re-run all of our optimization with a novel probe, optimizing the extraction of the topology rather than the distance of the target graph.
>
> We add the following discussion paragraph:
>
> &nbsp;&nbsp; L489: “Fourth, optimizing a probe to match pairwise distances might nevertheless be too restrictive.
> This approach might nevertheless be too restrictive. For example, the distance between “dog” and “river” does not have to be as precisely captured as the distance between “dog” and “cat”.
> Additionally, the exact choice of distance, for example, the shortest path distance on the WordNet graph, is itself questionable. To show that our approach can be generalized beyond distance learning, while simultaneously strengthening our results, we present supplementary results using a “contrastive” probe, which optimizes the extraction of the topology of the structure, irrespective of the distance.
> This probe is optimized according to the contrastive objective of Nickel et al 2017.
> On measures of topological fidelity (unlabeled undirected attachment score for syntax, rank for syntax and semantic), this probe achieved better scores than the distance probe, while revealing similar emergence dynamics.
> These results prove that our approach can be extended to different formalisations of the linguistic structure.”
>
> We support these claims with definition, methods and results in the appendix L1617 to 1693, Fig. H in appendix.
>
> *Reference: Nickel, M., & Kiela, D. (2017). Poincaré Embeddings for Learning Hierarchical Representations (arXiv:1705.08039). arXiv.*

---

> ### Author Response · Authors · 2025-11-20
> **Response to presentation issues**
>
> **Presentation 1**
> *"L257: "This hierarchical organization across layers highlights that these semantic embeddings are not present in the audio or text tokens’ embedding used as input to these transformer models. Instead, several computational steps are performed to merge and position these tokens’ embedding together into a semantic subspace." This really needs a lot of unpacking."*
>
> Layer-wise analyses are necessary to show that the linguistic structure indeed emerges after a certain number of processing. One expected result is that the layer at which the structure emerges should depend on its level in the linguistic hierarchy. Such findings have been obtained in the past with analyses of specific linguistic properties, and we reiterate them here.
>
> **Presentation 2**
> *"Why does the (inverse) U-shape indicate "hierarchical organization"?” “ And by semantic embeddings, I'm guessing what is intended is (purported) lexical semantic information?”
> Does the "Instead..." sentence offer anything more than just saying that there is layer-wise processing that contextualizes information with respect to other tokens in the input in a transformer?"*
>
>  The hierarchical organization here means that the linguistic structure is progressively extracted from the model (truly appearing only at middle layers) instead of existing in the model inputs.
> To make this statement clearer, we change the sentence in the following way:
>
> &nbsp;&nbsp; L271: “This is remarkable because transformer’s tokens are vectorised and could therefore contain some semantic information. Here, we show that such semantic information is not sufficient to instantiate the lexical semantic structure of nouns. Instead, several computational steps are performed to merge and position these tokens’ embedding together into a semantic subspace.“
>
> This is important, notably in light of Nickel et al. 2017 results, which demonstrate the possibility of embedding the WordNet tree.
> Consistent with this view, models of semantic code in the brain have independently turned to embeddings derived from WordNet, for example, Huth et al. 2016, instead of using simple semantic embeddings like Word2vec or the first layer of transformer models. Yet, these approaches do not follow a unified procedure across linguistic structures, which we propose here.
>
> *Reference: Nickel, M., & Kiela, D. (2017). Poincaré Embeddings for Learning Hierarchical Representations (arXiv:1705.08039). arXiv.; Huth, A. G., de Heer, W. A., Griffiths, T. L., Theunissen, F. E., & Gallant, J. L. (2016). Natural speech reveals the semantic maps that tile human cerebral cortex. Nature, 532(7600), 453‑458.*
>
> **Presentation 3**
> *"L316: "Language models are known to code syntactic trees through distances (Hewitt & Manning, 2019), while only correlates of syntactic code have been found in speech models (Pasad et al., 2024b).""*
>
> Previous approaches have measured if some statistics of the trees, like node depth, are decodable from the model activity, but these statistics are only correlated with certain dimensions of a vectorial representation coding the syntactic tree. To make this statement clearer, we rewrite it as:
>
> &nbsp;&nbsp; L308: "Language models are known to code syntactic trees through distances (Hewitt & Manning, 2019), while syntactic trees per se have yet to be found in speech models, since previous approach focused on decoding statistical quantities derived from the trees, like node depth (Pasad et al., 2024b).”
>
> **Presentation 4**
> *"What is the method with which the trees are reconstructed from the distance metric? (e.g., in Fig 4) Minimum spanning tree as in Hewitt and Manning? I'm also curious why UUAS was dropped as a metric for syntactic trees if the tree reconstruction was done."*
>
> We thank the reviewer for spotting that we forgot to mention this in the paper. The trees were indeed reconstructed using minimum spanning trees. We edit the legend of Fig. 4:
>
> &nbsp;&nbsp; “In each left subpanel, we plot an example of a 2d probe projection of the model activations, with trees reconstructed through the minimum spanning tree algorithm.”
>
> We computed the following metrics throughout our experiments: Spearman correlation, UUAS, and rank (Nickel et al. 2017). Spearman correlation was initially the only metric reported for the sake of simplicity. We now report UUAS and rank scores in the appendix (Fig. G in appendix), our results are unchanged when looking at these metrics. We add a paragraph in the appendix to define these metrics, L1170.
>
> *Reference: Nickel, M., & Kiela, D. (2017). Poincaré Embeddings for Learning Hierarchical Representations (arXiv:1705.08039). arXiv.*
>
> (The responses to presentation issues 5 to 7 are made in a separate comment, like the response to questions, soundness and representativeness issues).

---

> > ### Author Response · Authors · 2025-11-20
> > **Response to presentation issues 5-7**
> >
> > **Presentation 5**
> > *"L206: "This is considerably more than the 10^7 words that children experience in their first year on average." -> What is the source of this claim?"*
> >
> > We thank the reviewer for spotting that we missed the reference; these estimates come from all-day recording and are reported by Gilkerson et al. 2017. We add this reference line L206.
> >
> > *Reference: Gilkerson, J., Richards, J. A., Warren, S. F., Montgomery, J. K., Greenwood, C. R., Kimbrough Oller, D., Hansen, J. H. L., & Paul, T. D. (2017). Mapping the Early Language Environment Using All-Day Recordings and Automated Analysis. American Journal of Speech-Language Pathology, 26(2), 248‑265.*
> >
> > **Presentation 6**
> > *"L214: "Nevertheless, no analysis has shown the existence of a theoretically constructed semantic graph in a subspace of the model activity." -> First of all I'm not sure if I understand what exactly is in the set of "a theoretically constructed semantic graph". But there's plenty of representational studies based on information in semantic graphs and plenty on info from WordNet specifically (although they might not be looking at exactly the formulation of the problem looked at in this paper):
> > https://aclanthology.org/2023.findings-eacl.139
> > https://arxiv.org/abs/2402.16837
> > Maybe more importantly, I think Park et al. (2025) is currently cited after "the activation patterns of self-supervised models represent simple semantic relationships" but their analysis of hierarchical structure of words in WordNet in terms of shortest distance matrix between nouns seem exactly aligned in spirit with the lexical semantics part of this work. Surely there are differences but maybe this analysis at least merits a discussion?"*
> >
> > We thank the reviewer for these insightful references.
> > Indeed, Park et al. (2025) studied how WordNet subgraphs are represented in the ultimate layer of the model. They nevertheless employ a complementary approach, with results concentrated on a small subgraph of the WordNet graph (593 nouns, 364 verbs), while we focus on a larger subgraph (35352 nouns in total). The results of Park et al. focus on the last layers of the model, without explaining the internal code of the model.
> > To make the relationship between our work and the investigation in Park et al. (2025) clearer, we now correct the introductory sentence to the semantic results as follows:
> >
> > &nbsp;&nbsp; L255: “Additionally, models' behavior and the representation of their final layer have been shown to be consistent with an instantiation of lexical semantic relationship (Cohen et al. 2023), including the WordNet graph (Park et al 2025). To measure the internal layers’ representation, we train structural probes to predict the WordNet noun graph from text and speech models.”
> >
> > What we meant by “a theoretically constructed semantic graph” is a semantic graph proposed according to some linguistic theory of semantics. For example, the WordNet graph theorizes which synsets are linked through the hyponymy relationship.
> >
> > **Presentation 7**
> > *"I suggest using proper opening quotation marks in latex, all of them are currently closing quotation marks.
> > Figure 2D model identifiers are hard to map onto textual descriptions of the models for a reader who is not familiar with the identifiers. (e.g., I had to backtrace citations to figure out FMA is a music dataset, maybe relabeling them to match the descriptions in the text might help the readers)
> > L092: "Each level representation" -> "Each level of representation"
> > L118: made of text different text datasets -> made of different text datasets
> > L159: "score models with known strong cognitive abilities" seems like an unqualified claim. what does "strong cognitive abilities" mean?
> > L204: "Fig. 2D shows a gradual rise" -> 2E?
> > L256: "For this 200-D probe, the semantic scores across model layers follow a U-shape in all models" - Do you mean inverse U?"*
> >
> > We thank the reviewer for these suggestions and adopt the following editorial changes:
> >
> > &nbsp;&nbsp; Use of proper quotation marks in LaTeX.
> > &nbsp;&nbsp; Simplification of Fig 2D model identifiers.
> > &nbsp;&nbsp; correction of lines L092, L118, L204, L256.
> >
> > Concerning the statement that Llama2 has “strong cognitive ability”, we meant that these models have stronger linguistic competence than Pythia models.
> > We correct our phrasing and refer to a study that precisely demonstrated this (Waldis et al 2024).
> >
> > &nbsp;&nbsp; L167 “The two Llama 2 models, of size 7B and 13B, allowed us to score models with stronger linguistic scores (Waldis et al. 2024), defining an upper-bound for what the smaller Pythia models could achieve.”
> >
> > *Reference: Waldis, A., Perlitz, Y., Choshen, L., Hou, Y., & Gurevych, I. (2024). Holmes.  A Benchmark to Assess the Linguistic Competence of Language Models. Transactions of the Association for Computational Linguistics, 12, 1616‑1647.*

---

> ### Author Response · Authors · 2025-11-20
> **Response to questions**
>
> **Question 1**
> *"If there are any responses to the soundness issues I raised, I'd be happy to engage! The main claim didn't feel fully justifiable to me but I'd be happy to update my beliefs if there is good justification."*
>
> We look forward to engaging in a constructive set of review rounds with the reviewer, and these remarks have already been very insightful.
>
> **Question 2**
> *"Is it problematic that the phonemic scores for the models trained with nonlinguistic data like music is very high? It being high for French models is understandable, since some vowel distinctions in English will be shared in French vowels, but it feels like a Spearman correlation of >0.5 is almost excessively high for a model trained on music or environmental sounds."*
>
> Unfortunately, the training database of these models is not entirely void of any speech, as measured in Orhan et al. 2025. This is an important caveat of these control models and could explain the high score already reached by control models. We add the following sentence in the phoneme section:
>
> &nbsp;&nbsp; L207 “Note also, that the music and environmental sounds models are not completely naive to speech, with an estimated 8\% of speech inputs in the environmental sound dataset (Orhan et al. 2025)”
>
> These control models have nevertheless been shown to have reduced performance on a Logatome classification task in Orhan et al. 2025, compared to the English-exposed models. For these reasons, we believe that these scores are indeed high estimates compared to scores a model trained without any speech would likely reach.
>
> *Reference: Orhan, P., Boubenec, Y., & King, J.-R. (2025). The detection of algebraic auditory structures emerges with self-supervised learning. PLOS Computational Biology, 21(9), e1013271.*
>
> **Question 3**
> *Given the discussion about linear trees throwing off the metrics on p7 (which I appreciated), maybe it would be good to put the correlation with linear tree directly on Fig 4D as another baseline, eg as a horizontal dotted line on the graph.*
>
> We thank the reviewer for stressing the importance of this linear metric, which is indeed key to showing the improvement over the baseline. We can't put a horizontal dotted line on the graph as the correlation with the linear tree change for each model. We could, nevertheless, plot in the appendix a new figure that quantifies the difference between the linear tree and the syntactic tree for each model if the reviewer would like to see quantitative results of this effect.

---

### Official Review · Reviewer_4wAq · 2025-11-01

**Soundness:** 3
**Presentation:** 3
**Contribution:** 2
**Rating:** 6
**Confidence:** 3

**Summary:**

- The paper investigates whether semantic categories emerge in transformer-based text and audio models during training and make comparisons with WordNet structured semantic graph. Among other things, the authors evaluate when different semantic categories (e.g., "mammal", "dog") emerge in the models by probing representations at different training checkpoints. Authors do not observe a sequential emergance of semantic categories, but observe a sequential emergence across inguistic structures. The analysis is done across various models (Pythia, Llama) and includes both text and audio models.
- The paper is well-structured but some parts are dense.
- The assumption that linguistic structures are encoded in a geometric code via linear subspaces is a simplification, but it is supported by prior work showing that linear probes can extract meaningful syntactic and semantic structures. While the brain and neural networks likely use more complex mechanisms, this approach provides a tractable and interpretable way to study the emergence of linguistic representations during training. Thus, the methodology is sound but not new.

**Strengths:**

- The papers is well furnished in terms of references to the litterature.
- Construction of a paired text-audio benchmarks with a great amount of data is welcome.
- Use of various models (Pythia, Llama2) and audio models (Wave2vec2).

**Weaknesses:**

- Assuming linear and distance matrices for language representations can be considered as too simple way nowadays to explore representations in NN.
- The method uses knn classification to score how well categories are separated. The use of knn is simple and may miss fine details.
- It is not clear if retrieving the geometry of phonemix articulation is an impressive result, given that 20 years old models (e.g. Oudeyer) where already able to represent such geometry with Self-Organising Maps.
- The paper is well-structured but some parts are dense. Authors also make some shortcuts that don't always seemed justified. For instance, the "double bump" in fig 2C is rather flat in-between 0.5 and 0.8 relative depth, which is know to be the most expressive depth for various analysis such as brain encoding. Thus, this sentence don't seem justified: "This demonstrates that instantiating
the phoneme structure is not performed trivially with random acoustic processing."
- Only small random models where explored (<= 10^8 params), e.g. Fig 3D. Why? As they don't need training, it should be rather quick to test to see if some effect emerge at 10^10 parameters.

**Questions:**

- I understand that authors want to find clear linear relations, but why other methods, even unsupervised method already estabished since 5 years (e.g. UMAP), where not used to explore relative distances in projected space?
- Only small random models where explored (<= 10^8 params), e.g. Fig 3D. Why?
- How do these results relate to in-context learning or prompting? Could such "linear analyses" performed in such context?

---

> ### Author Response · Authors · 2025-11-20
> **Response to questions**
>
> We would like to thank the reviewer for the detailed comments, and note that the reviewer mentions a  “well-structured” and “furnished paper in terms of reference”, using a “sound” although “not new” methodology.
>
> **Question 1**
> *“I understand that authors want to find clear linear relations, but why other methods, even unsupervised method already estabished since 5 years (e.g. UMAP), where not used to explore relative distances in projected space?”*
>
> We now modified the manuscript to clarify this issue:
>
> &nbsp;&nbsp; L100 “Following this established interpretability paradigm, we restrict our focus to probing of representations can be directly used by downstream neurons through a simple linear readout. In contrast, non-linear probing could recover the target distances from arbitrary representations, thus hindering interpretability.”
>
> We add a supplementary figure (Fig D in appendix) that uses UMAP to visualise the probed space before and after projection, and the following paragraph in the appendix:
>
> &nbsp;&nbsp; L1161: “We plot two UMAP visualisations before and after projection with the 200-D probe. Before projection, the space is only roughly divided into large semantic categories, presenting many small clusters without a clear organisation. After projection, the space is well organised into continuous clusters mirroring the WordNet tree hierarchy.
> Consequently, this unsupervised visualisation demonstrates that the WordNet tree is only instantiated in a subspace of the model activity, which is found by our probe. “
>
>
> **Question 2**
> *"Only small random models where explored (<= 10^8 params), e.g. Fig 3D. Why? As they don't need training, it should be rather quick to test to see if some effect emerges at 10^10 parameters."*
>
> We added the evaluation of two randomly initialised transformer models with 13B and 1B parameters (Fig 3 and 4). Consistent with results for speech models, no instantiation of semantic structures (max semantic score: 0.008) or syntactic structures (max syntactic score: 0.56) is observed in these models.
>
> **Question 3**
> *"How do these results relate to in-context learning or prompting? Could such "linear analyses" performed in such context?"*
>
> We would like to thank the reviewer for pointing out the possibility of applying our research to in-context learning or prompting. This is a good point. We now indicate:
>
> &nbsp;&nbsp; L514: “Our probing approach could be used to understand how structures are built through in-context learning. For example, (Orhan et al. 2025) show that algebraic auditory structures are detected by speech models through in-context learning. This work predicts that the code underlying this detection is a distance code which can be detected through a linear probing approach.”
>
> In the case of prompting, the relationship is less clear unless the prompt introduces a clear structure.
>
> *Reference: Orhan, P., Boubenec, Y., & King, J.-R. (2025). The detection of algebraic auditory structures emerges with self-supervised learning. PLOS Computational Biology, 21(9), e1013271.*
>
> (The responses to weaknesses are given in a separate comment.)

---

> ### Author Response · Authors · 2025-11-20
> **Response to weaknesses**
>
> **Weakness 1**
> *"Assuming linear and distance matrices for language representations can be considered as too simple way nowadays to explore representations in NN. The method uses knn classification to score how well categories are separated. The use of knn is simple and may miss fine details."*
>
>
> We agree with the reviewer that KNN classification might not fully capture the extent to which semantic categories are or aren’t captured in the model. This decision is motivated by a compute constraint:  each classification requires resolving a high-dimensional classification on a dataset of 35352 points for each model, for each checkpoint and for each of the 35352 semantic categories.
>
> **Weakness 2**
> *"It is not clear if retrieving the geometry of phonemix articulation is an impressive result, given that 20 years old models (e.g. Oudeyer) where already able to represent such geometry with Self-Organising Maps."*
>
> We agree with the reviewer that the geometry of phonemic articulation, especially of vowels, is ultimately dictated by a few articulatory features which are relatively easy to learn in a controlled auditory environment, as found by Oudeyer and colleagues.
> Nevertheless, we show that contrary to these models, deep neural networks learn to instantiate several linguistic structures in distinct subspaces, which provide a unifying framework to study the emergence of linguistic structures.
>
> **Weakness 3**
> *"The paper is well-structured but some parts are dense. Authors also make some shortcuts that don't always seemed justified. For instance, the "double bump" in fig 2C is rather flat in-between 0.5 and 0.8 relative depth, which is know to be the most expressive depth for various analysis such as brain encoding. Thus, this sentence don't seem justified: "This demonstrates that instantiating the phoneme structure is not performed trivially with random acoustic processing."*
>
> We agree that the double bump is rather flat between 0.5 and 0.8 depth. We correct this sentence:
>
> &nbsp;&nbsp; L191 “We find that phonemes are best encoded in layers with a relative depth of 0.5 and 0.8 (Fig. 2C), achieving a Spearman correlation of 0.75. This phonemic score outperforms by a large margin the untrained, random, baseline (phonemic score of 0.2). This demonstrates that instantiating the phoneme structure is not performed trivially with random acoustic processing.”
>
> (The responses to questions are given in a separate comment.)

---

### Comment · Area_Chair_Bn59 · 2025-11-25
**Please discuss**

This paper has a wide range of scores. I would love to see the reviewers engage with each other, as well as the author response, and see if they come closer to a consensus.  Have the rebuttals addressed your concerns or clarified anything?

---

### Author Response · Authors · 2025-12-01
**Summary Comment**

We here summarise the rebuttal discussions for the Area Chair.

**Summary**
Reviewers provided a wide range of scores: 10, 2, 4, 6.
Overall, we addressed the reviewer concerns by showing significance when unclear, demonstrating the robustness of the work when changing the fundamental hypothesis of distance-based representation, and running a series of novel experiments on single-unit analysis and visualisations. The paper's results were not changed by these discussions, but several interesting experiments were added in the appendix, increasing the robustness of the finding.

**Summary of each response**


The highest-grading reviewer stressed with confidence the “importance”, “rigour”, “clarity” and “difficulty” of the work and proposed two follow-up experiments.
The first concerned the emergence curve per layer; we ran this experiment and reported a slight difference in emergence between early and late layers.
The second aims to split the emergence into more fine-grained syntactic categories, and the reviewer suggested that this could be the focus of future work. We agreed and responded by proposing a protocol based on existing literature.

The lowest grading reviewer stressed an “interesting” and “relevant” work, but with soundness and representativeness issues.
Soundness issues questioned whether the semantic structure was significantly instantiated by the speech models. The reviewer stressed that a weak instantiation is not sufficient to then make a claim of a sequence: phoneme, semantic, syntax, and to further extend this claim to a claim of similarity with children's learning dynamics.
We responded to this issue with a quantitative proof of the significance of the semantic structure instantiation.
As pointed out by the reviewer, our discussion already stressed that this finding should be taken with care despite its significance, and we clarified every sentence that would imply a sequence of emergence, instead stressing two stages: phonemic, followed by lexical-syntactic emergence.

Representativeness issues first questioned why the probing was focused on a certain part of speech, and second questioned the use of the distance probe despite its importance in the literature, asking if other formalisations could not be proposed.
We responded to the first issue by clarifying a methodological confound that could arise with probing all parts of speech simultaneously.
To the second issue, we agreed with the reviewer that a distance-based representation might not be the only solution to the instantiation of linguistic structure. Although this could be the scope of future work, we re-ran the paper experiments on syntax and semantics with a novel representation proposal, which optimises the topology instead of the distance, through a contrastive approach. This led to similar results and stressed the robustness of the work while opening a new approach to be explored in future works.
We additionally answered all questions and presentation points of the reviewer by providing clarified experimental results, including alternative metrics.

The reviewer with grade 4 stressed the “interesting” and “systematic” aspect of the work with “carefully designed” probes, despite two weakness points.
The first concerned the absence of phoneme analysis by text models, and proposed an experiment that we ran, showing that these models do not perform the instantiation of articulatory features from a sequence of IPA symbols, as expected.
The second weakness concerned the absence of single-unit analysis and asked for clarification of the correlation between subspace. We filled this weakness with 3 novel sets of measurements reported in the appendix, showing the independence of these subspaces and the difference between how semantic and syntactic subspaces are distributed over single units.

The reviewer with grade 6 “welcomed” a “well-structured” paper with a “sound” methodology, but questioned if the linear supervised probing approach was not too simple compared to the unsupervised approach.
To answer this question, we showed, on semantics, that a non-linear, unsupervised, dimensionality reduction technique initially failed before probing, but then succeeded after the linear approach was applied. This demonstrates the difficulty of these measurements and the advantage of a first supervised step, while reconciling it with an unsupervised approach, which can be applied for visualisation as a second step.
At the request of the reviewer, we also extended the measurement to other randomly initialised networks, which did not change the finding.

Despite the shortening of the rebuttal period this year, we would like to thank the reviewers for their feedback and the improvements it led to in this work.

---

### Meta-Review · Area_Chair_BjEH · 2025-12-11

**Summary:**

While the topic and findings seem interesting and exciting, to the extent some recommend it for a prize. Many soundness issues, comments on the claims and details of the work were raised. A high variand is usually a good sign for a paper that is exciting and with potential, but also the many issues suggest to me that this is not ready yet and hopefully it will be wonderful, when it is.

**Reviewer Concerns:**

Many concerns on "soundness" as one especially careful reviewer notes exist beyond the more minor ones.

**Reviewer Scores:**

That is not a fair, relevant or meaningful question. I protest the way this was all handled.
A Reviewers are not here, and ToM is weak, at least mine and the one literature study. I will not try to predict people.
B Scores are, anyway, a weak signal of interest; a paper should not be accepted or rejected just based on it. An AC's job is to look at the specific weaknesses and translate them into a recommendation.
C There are about 100 pages of discussions for me to read overall, in addition to the discussions I monitored and were just replaced, this is beyond my personal ability to do fairly. I did my best effort.

---

### Decision · Program_Chairs · 2026-01-26

Reject